# Human Disturbance Increases Health Risks to Golden Snub-Nosed Monkeys and the Transfer Risk of Pathogenic Antibiotic-Resistant Bacteria from Golden Snub-Nosed Monkeys to Humans

**DOI:** 10.3390/ani13193083

**Published:** 2023-10-03

**Authors:** Shuzhen Zou, Tingting Yuan, Tan Lu, Jiayu Yan, Di Kang, Dayong Li

**Affiliations:** 1Key Laboratory of Southwest China Wildlife Resources Conservation of Ministry of Education, China West Normal University, 1# Shida Road, Nanchong 637009, China; zousz@foxmail.com (S.Z.); kangyuyao@foxmal.com (D.K.); 2Key Laboratory of Conservation Biology of Rhinopithecus Roxellana at China West Normal University of Sichuan Province, China West Normal University, 1# Shida Road, Nanchong 637009, China; 3Liziping Giant Panda’s Ecology and Conservation Observation and Research Station of Sichuan Province, Science and Technology Department of Sichuan Province, Chengdu 611233, China

**Keywords:** antibiotic resistance genes, eco–environment coupling system, sentinel animal, drug-resistant pathogen, environmental health assessment, *Rhinopithecus roxellana*

## Abstract

**Simple Summary:**

The conflict between humans and wild animals is an inevitably negative product of human disturbance (HD). There is a transfer of antibiotic resistance genes (ARGs) among wild animals, domestic animals, humans, and their food. On the one hand, if ARGs enter from humans to wild animals, they will have unknown effects on the wild animals. On the other hand, if pathogenic bacteria carrying ARGs in wild animals are transferred to humans, this might bring new diseases to humans. In this report, golden snub-nosed monkeys (*Rhinopithecus roxellana*) were used as sentinel animals to test the effect of HD on the health risk of wild animals and the transfer risk of pathogenic antibiotic-resistant bacteria (PARBs) from wild animals to humans. Our result could provide a method to evaluate the effect of HD on the health risk of wild animals and transfer risk of PARBs from wild animals to humans under the eco–environment coupling system of wild animals and humans. In the future, the conflicts between wild animals and humans should be reduced.

**Abstract:**

From the perspective of interactions in the human–animal–ecosystem, the study and control of pathogenic bacteria that can cause disease in animals and humans is the core content of “One Health”. In order to test the effect of human disturbance (HD) on the health risk of pathogenic antibiotic-resistant bacteria (PARBs) to wild animals and transfer risk of the PARBs from wild animals to humans, golden snub-nosed monkeys (*Rhinopithecus roxellana*) were used as sentinel animals. Metagenomic analysis was used to analyze the characteristics of PARBs in the gut microbiota of golden snub-nosed monkeys. Then, the total contribution of antibiotic resistance genes (ARGs) and virulence factors (VFs) of the PARBs were used to assess the health risk of PARBs to golden snub-nosed monkeys, and the antimicrobial drug resistance and bacterial infectious disease of PARBs were determined to assess the transfer risk of PARBs from golden snub-nosed monkeys to humans. There were 18 and 5 kinds of PARBs in the gut microbiota of golden snub-nosed monkeys under HD (HD group) and wild habitat environments (W group), respectively. The total health risks of PARBs to the W group and the HD group were −28.5 × 10^−3^ and 125.8 × 10^−3^, respectively. There were 12 and 16 kinds of KEGG pathways of human diseases in the PARBs of the W group and the HD group, respectively, and the gene numbers of KEGG pathways in the HD group were higher than those in the W group. HD increased the pathogenicity of PARBs to golden snub-nosed monkeys, and the PARBs in golden snub-nosed monkeys exhibited resistance to lincosamide, aminoglycoside, and streptogramin antibiotics. If these PARBs transfer from golden snub-nosed monkeys to humans, then humans may acquire symptoms of pathogens including *Tubercle bacillus*, *Staphylococcus*, *Streptococcus*, *Yersinia*, *Pertussis*, and *Vibrio cholera*.

## 1. Introduction

Wild animal health and habitat health are affected by human activities [1,2]. To prevent the effect of human disturbance (HD) on wild animals, China has invested resources in the construction of wild animal nature reserves in their habitats and achieved good results [3]. However, the fundamental contradiction between the social and economic development of humans and the protection of wild animals in wild animal nature reserves is difficult to reconcile. Moreover, HD, such as grazing, planting, rural domestic wastewater, household waste, and ecological tourism, encompass factors that affect the survival of wild animals in wild animal nature reserves [4]. Thus, there are large areas that suffer from HD in wild animal nature reserves in China. The conflict between humans and wild animals is an inevitably negative product of HD [3].

Antibiotic resistance genes (ARGs) are a quintessential issue in “One Health” and pose the most serious threat to environmental health [5]. There is a transfer of ARGs among wild animals, domestic animals, humans, and their food in the habitats of wild animals under HD [6]. The gut microbiota is the “second genome” of wild animals and plays a key role in the physiological and biochemical processes of wild animals through coevolution with wild animals [7]. If ARGs associated with HD enter the gut of wild animals, they will have unknown effects on the “second genome” of wild animals [8]. More importantly, the adaptive evolution of pathogenic bacteria caused by ARGs may increase the capacity of virulence factors (VFs), which leads to the undesirable situation of omnidirectional mutations in pathogenic bacteria [9]. When ARGs and VFs coexist in one bacterium, the bacterium is a pathogenic antibiotic-resistant bacterium (PARB). PARBs can be used to evaluate the health risk of HD in the ecological environment [10,11]. Mobile genetic elements (MGEs) can carry ARGs or VFs of pathogenic bacteria, allowing their diffusion into the environment [10]. If the PARBs in wild animals transfer over to humans, they may infect humans and cause emergencies [12]. 

Sentinel species are a group of animals used in a given area to monitor or warn of environmental pollution caused by toxic or potentially toxic substances [13]. Selecting wild animal species to be used as sentinels to assess the health risk of PARBs to wild animals and the transfer risk of PARBs from wildlife to humans is challenging [14]. To conduct research on golden snub-nosed monkeys (*Rhinopithecus roxellana*), some have been confined in their wild habitat, where they are highly regulated by humans, so their feces are easy to sample and manipulate. Studies in these areas have shown that there are good and visible changes in intestinal microbes [15,16]. Peng et al. [17] showed that HD could make the gut microbiota of golden snub-nosed monkeys more similar to that of humans, and Yu et al. [18] indicated that the gut microbiota of golden snub-nosed monkeys could be used as a key indicator of their health. However, only a few studies focused on the health risks of PARBs to golden snub-nosed monkeys and the transfer risk of PARBs from golden snub-nosed monkeys to humans.

In this study, the characteristics of PARBs in the gut microbiota of golden snub-nosed monkeys were analyzed with metagenomic sequencing, which was used to study the effect of HD on the health risks of PARBs to wild animals and the transfer risk of PARBs from wild animals to humans. Our results showed that HD increased the pathogenicity of PARBs in wild animals, and PARBs in wild animals were resistant to lincosamide, aminoglycoside, and streptogramin antibiotics. If these PARBs transfer from golden snub-nosed monkeys to humans, then humans may acquire symptoms of pathogens including *Tubercle bacillus*, *Staphylococcus*, *Streptococcus*, *Yersinia*, *Pertussis*, and *Vibrio cholera*. Our results could assist with developing the management and treatment of such diseases in wild animals and emergency measures for the transfer of pathogens to humans.

## 2. Research Methods

### 2.1. Study Subjects and Samples Collection

The golden snub-nosed monkey species is a rare and endangered primate in China. The Baihe National Nature Reserve has the highest density and largest and most representative population of golden snub-nosed monkeys. The home ranges of golden snub-nosed monkeys in Baihe National Nature Reserve are marked with light green in Figure 1. Previous studies found that grazing in the reserve is serious, and the HD trace points, especially grazing trace points, overlap with the home ranges of golden snub-nosed monkeys [4]. The home range of some golden snub-nosed monkey families in Xiapingdi has been severely disturbed by grazing [4]. These families were confined in their home range in 2021 for ecotourism and scientific research; thus, they have been disturbed by humans (HD group) [16]. The other golden snub-nosed monkeys within the Xiapingdi home range live in mountains that are inaccessible, and they never are disturbed by humans (W group). There was an HD group and a W group in the Xiapingdi home range, so these groups of Xiapingdi golden snub-nosed monkeys were chosen as the study subjects in this study. 

A total of 50 and 53 fresh feces samples from golden snub-nosed monkeys were randomly collected from the HD group and the W group, respectively. The feces were collected at 9 am and 5 pm every day. The methods for collecting fecal samples are reported in Guo et al. [19]. All fecal samples were transported to the Shanghai Magi Biological Company on dry ice for 16S rRNA and metagenomic sequencing after collection to ensure the freshness of samples from the W and HD groups.

### 2.2. DNA Extraction

A FastDNA Spin Kit (MP Biomedicals, France) was used to extract DNA from 0.5 g of fresh golden snub-nosed monkey feces from the HD group and the W group following the manufacturer’s instructions. Subsequently, spectrophotometric analysis (Nanodrop ND1000, USA) and 1.0% agarose gel electrophoresis were performed to assess the quality and concentration of isolated DNA. Finally, the isolated DNA was stored at −80 °C for further analysis. The DNA extraction was performed by the lab of Shanghai Magi Biological Company, China (Meiji Biological Medicine Co., Ltd., Shanghai, China). 

### 2.3. Analysis of 16S rRNA and Metagenomic Sequencing 

The V3-V4 hypervariable regions of 16S rRNA genes were amplified with PCR using the barcoded primer set 338F (5′-ACTCCTACGGGAGGCAGCA-3′) and 806R (5′-GGACTACCAGGGTATCTAAT-3′) [20,21]. In our study, a total of 103 samples labeled with different barcodes of equal DNA content were pooled into a library for sequencing. Amplicon sequencing was performed using a MiSeq 300 instrument (Meiji Biological Medicine Co., Ltd., Shang hai, China). The data were deposited into the NCBI Sequence Read Archive (SRA) database under accession number PRJNA947971. The data quality control and stitching methods were performed as reported by Zhu et al. [22]. Before a downstream analysis, OTU clustering was performed for non-repeating sequences (excluding single sequences) according to a 97% similarity [23]. The chimera was removed during the clustering process to obtain representative sequences of OTUs; subsequently, all optimized sequences were mapped to the OTU representative sequences, and selected sequences with more than 97% similarity to the representative sequences were used to generate OTU tables. 

Representative samples were selected according to the β diversity of gut microbiota to obtain all information on the gut microbiota of golden snub-nosed monkeys. According to the results of the 16S rRNA sequencing, 15 and 15 representative DNA samples of gut microbiota were selected from the 50 and 53 DNA samples for metagenomic sequencing analysis of the W group and HD group, respectively, following the selection criterion of “most representative” using the micro-PITA method [24]. The representative samples are shown in Appendix A. We used Covaris M220 (Gene Company Limited, Shanghai, China) to fragment the obtained DNA and subsequently screened fragments of approximately 400 bp to construct the paired-end library using NEXTFLEX Rapid DNA-Seq according to the manufacturer’s instructions. HiSeq 2000 (Illumina Inc., San Diego, CA, USA) was selected for metagenomic sequencing of the paired-end library at Majorbio Bio-Pharm Technology Co., Ltd. (Shanghai, China). The data were deposited into the NCBI Sequence Read Archive (SRA) database under accession number PRJNA947945. For the obtained metagenomic data, we used fastp to perform quality clipping of adapter sequences from reads and remove low-quality reads (reads less than 50 bp in length with an average mass value of less than 20 and containing N bases) [25]. The reads were compared with the host genome sequence using BWA software (https://bio-bwa.sourceforge.net). Reads with high similarity were removed to obtain clean reads for subsequent analysis. Next, clean reads were assembled using Megahit for contigs, and contigs with the shortest sequence longer than 300 bp were retained [26]. We used MetaGene to perform an open reading frame prediction of the retained contigs (length of reads ≥ 100 bp), and the per-base coverage depth across all contigs was calculated by mapping raw reads from each sample [27]. CD-HIT was used to construct the coverage >90% and identity >90% of predicted gene sequences of all samples as a nonredundant gene contig. The identity ≥ 0.95 of nonredundant gene contig was blasted using SOAPaligner, and gene abundance of the nonredundant gene contigs was subsequently calculated. Then, the nonredundant gene contig was predicted as the open reading frame (ORF), and the ORFs were used in card_v3.0.9 (the Comprehensive Antibiotic Research Database) to obtain ARGs (http://arpcard.mcmaster.ca, (accessed on 28 November 2022); blastp, e-value: 1 × 10^−5^). Vfdb setB (http://www.mgc.ac.cn/VFs/) was used to obtain VFs (blastp, e-value: 1 × 10^−5^), and plasmids in ACLAM were used to obtain MGEs (blastp, e-value: 1 × 10^−5^). An identity > 80% of ARGs, VFs, and MGEs was chosen for analysis.

To compare coverage between different samples, the coverage of ARG-like ORFs and VF-like ORFs was normalized using the data size of each sample (copies/Gb) [28,29].

### 2.4. Assembly of Metagenome Assembly Genomes (MAGs)

The filtered clean reads were grouped by sampling W group and HD group and co-assembled using MEGAHIT v1.13 with default parameters [30]. These co-assembled contigs were clustered to recover metagenomes using metaBAT (contig length ≥ 1000 bp) [31]. The assembled bacterial genomes were further refined to produce high-quality individual genomes using the built-in refining module of MetaWRAP, with the selection criteria of >50% completeness and <5% contamination. Following that procedure, all metagenome data were refined to remove redundant assemblages and then annotated for taxonomic classifications using the Genome Taxonomy Database (GTDB; v1.4.0) [32]. Next, high-quality bins were reassembled to reconstruct the MAGs using SPAdes [33]. The assembled genomes of MAGs were used to blast the database of ARGs (blastp, evalue: 1 × 10^−5^, identity > 70%, http://arpcard.mcmaster.ca) and the database of VFs (blastp, evalue: 1 × 10^−5^, identity > 70%, http://www.mgc.ac.cn/VFs/) to obtain PARBs [34]. Then, the assembled genomes of PARBs were used to blast the database of KEGG pathways (blastp, evalue: 1 × 10^−5^, identity > 70%, https://www.genome.jp/kegg) and the database of MEGs (blastp, evalue: 1 × 10^−5^, identity > 70%, http://aclame.ulb.ac.be/). The coverage of each MAG was calculated as the average scaffold coverage, and each scaffold was weighed by its length in base pairs. Then, the relative abundance of each MAG was calculated as its coverage divided by the total coverage of all genome bins.

Thus, the relative abundance of each MAG was calculated as the number of reads (based on average coverage) aligning to the MAG normalized by the total number of reads in the sample. The calculation formula was as follows:(1)$$A_{i}=C_{i}\times (N_{i}/N)$$
where A_i_, C_i_, and N_i_ represent the relative abundances, coverage, and contig numbers of i MAG, respectively. N is the total number of reads in the samples.

### 2.5. Health Risk of PARBs to Golden Snub-Nosed Monkeys

The subtypes of ARGs, the numbers of ARGs, the subtypes of VFs, and the numbers of VFs in each PARB in the W group and HD group were four independent variables. We counted the subtypes of ARGs, the gene numbers of ARGs, the subtypes of VFs, and the gene numbers of VFs in each PARB in the W group and HD group, respectively, and the principal component features of the four independent variables were obtained using factor analysis. Then, the total scores of the principal component features were calculated using Formula (2). The greater the score of the four independent variables, the higher the contribution of ARGs and VFs to the risk of one PARB. The health risk of one PARB was affected by its abundance, so the health risk of one PARB was calculated using Formula (3), and the total health risk of all PARBs in the W group and HD group was calculated using Formula (4) [35].
(2)$$F_{i}=\sum_{1}^{i}S_{i}\times W_{i}$$
(3)$${HR}_{i}=F_{i}\times A_{i}$$
(4)$$THR=\sum_{1}^{n}{HR}_{i}$$
where i is the number of component features in the W group or the HD group, *S_i_* is the score of the component features, W*_i_* is the weight of *S_i_* and F_i_ is the total score of the PARB. HR*_i_*is the health risk of one PARB, and THR is the total health risk of all PARBs in the W group or the HD group. The greater the total health risk, the greater the health risk of PARBs to golden snub-nosed monkeys [35]. 

Finally, KEGG pathways in antimicrobial drug resistance and bacterial infectious disease of PARBs were determined to assess the transfer risk of PARBs from golden snub-nosed monkeys to humans. The conceptual framework of the health risk assessment for wild animals and humans is shown in Figure 2. 

### 2.6. Research Techniques

The abundance of the gut microbiota in golden snub-nosed monkeys at the OTU level was used to calculate the Simpson’s diversities and enterotypes of the gut microbiota. The relative abundance of ARGs at the subtype level was used to calculate their Simpson’s diversities using the R package “vegan” [36]. The core genes of MAGs were used to construct a phylogenetic tree using the maximum likelihood (ML) method, and the phylogenetic tree was visualized with Mega 7.0. The diversities of gut microbiota and ARGs, as well as the gene numbers of pathogenic ARGs, MGEs, and KEGG pathways, were calculated as their mean values ± standard error in the HD group and the W group, and their difference between the HD group and the W group were analyzed using a *t*-test with a significance level of *p* < 0.05. Origin 8.5 was used to produce other graphics in this study, and the analysis of variances was performed using SPSS software 2021.

## 3. Results

### 3.1. The Effect of HD on ARGs and VFs in the Gut Microbiota of Golden Snub-Nosed Monkeys

Figure 3A shows that the Simpson diversity indices of gut microbiota in the HD group were significantly lower than those in the W group (*p* < 0.001). There were two types of enterotypes in the gut microbiota of golden snub-nosed monkeys, the gut microbiota of all samples in the HD group were type 1, while those in the W group were type 2 (Figure 3B). There were 77 subtypes of ARGs that were classified into 13 classes of ARGs in both the HD group and W group (Figure 3C,E), and HD significantly increased the relative abundances of ARGs in multidrug, tetracycline, MLS, and glycopeptide, as well as mupirocin and nucleoside types that were checked not only in the W group but also in the HD group. The Simpson diversity indices of the ARGs were increased by HD (Figure 3D), and Table 1 shows that the contig numbers of pathogenic ARGs were also increased by HD.

### 3.2. The Effect of HD on PARBs in the Gut Microbiota of Golden Snub-Nosed Monkeys

Our results showed that 114 and 54 high-quality MAGs were identified from whole metagenomes (Appendix A). Among them, 18 and 5 MAGs were PARBs in the HD group and W group, respectively, and the total relative abundance of PARBs in the HD group was higher than that in the W group (Figure 4A,B). The PARBs belonged to Firmicutes, Actinobacteria, and unclassified bacteria in the HD group, while the PARBs belonged to Firmicutes and unclassified bacteria in the W group. Further analysis found that the PARBs carried four types of ARGs and eleven types of VFs in the HD group, while those in the W group carried two types of ARGs and six types of VFs. Eight species of PARBs in the gut microbiota of the HD group were unclassified, while there was only one species of PARBs in the gut microbiota of the W group. In the HD group, 12 species of MAGs were multidrug PARBs, while 4 species of MAGs were multidrug PARBs in the W group. 

### 3.3. The Effect of HD on Health Risks to Golden Snub-Nosed Monkeys

In our study, factor analysis was used to calculate the contribution of ARGs and VFs to PARBs (Appendix A, *p* = 0.000002). The results showed that the contributions of F_1_, F_3_, and F_5_ to health risk were controlled by VFs, while the contributions of F_4_ and F_6_ were controlled by ARGs, and that of F_2_ was controlled by VFs and ARGs. The total contribution of F_1_, F_3,_ and F_5_ to the risk of PARBs was 48.581%, while that of F_4_ and F_6_ to the risk of PARBs was 17.091%. Table 2 showed that the total score of PARBs in the HD group was 1.11, while that in the W group was −1.11. The relative abundance of PARBs in the HD group was 873.1 × 10^−3^, while that in the W group was 126.8 × 10^−3^. Further analysis found that the total health risks of PARBs to the W group and the HD group were −28.5 × 10^−3^ and 125.8 × 10^−3^, respectively. 

### 3.4. The Effect of HD on the Transfer Risk of PARBs from Golden Snub-Nosed Monkeys to Humans

Figure 5A shows that the total gene numbers of MGEs were 40 and 186 in the PARBs of the W group and the HD group, respectively. Figure 5B,C shows that the total gene numbers of KEGG pathways in bacterial infectious diseases and antimicrobial drug resistance in the HD group were higher than in the W group (*p* < 0.05; *p* < 0.01). The KEGG pathways of bacterial infectious disease that were only found in PARBs in the HD group included ko05152, ko05100, ko05133, and ko05110. We found that the PARBs could produce antimicrobial drug resistance to antibiotics by KEGG pathways including ko01523, ko01524, ko01501, ko01502, and ko01503. The hosts of additional KEGG pathways in bacterial infectious disease were G_bin134 (unclassified bacteria), G_bin221 (Actinobacteria), G_bin485 (Actinobacteria), and G_bin645 (Firmicutes). 

## 4. Discussion

If the diversity of gut microbiota decreases, the stability of gut microbiota should also decrease [37]. Considering our results, this suggests that HD reduced the stability of the gut microbiota in golden snub-nosed monkeys. In general, enterotypes are stable, but if there are long-term changes in dietary habits or other strong disturbing factors to the host, enterotypes may change [38]. Our study indicated that the enterotypes of the gut microbiota of golden snub-nosed monkeys were changed because the stability of the gut microbiota was reduced. Some diseases of a host are related to gut microbiota enterotypes in the host [38]. Thus, the health of the studied golden snub-nosed monkeys may have been altered by HD. The health of humans and animals is being threatened by ARGs under the eco–environment coupling system of wild animals and humans [39]. The ARGs of gut microbiota in the golden snub-nosed monkeys were important indicators of their health management for conservation [40]. The higher the diversity of ARGs, the greater the risk of ARGs [41]. If both an ARG and a VF are present in one ORF, the ARG is a pathogenic ARG [10]. The ecological risk of ARGs is also related to the pathogenicity of the pathogenic ARGs, and the pathogenic ARGs reflect the highest risk level of ARGs [42]. The Simpson diversity indices of ARGs and the gene numbers of pathogenic ARGs in the gut microbiota of golden snub-nosed monkeys indicated that the risk of ARGs to gut microbiota was enhanced by HD. Our published studies found that the relative abundances of ARGs and pathogenic bacteria in the gut microbiota of golden snub-nosed monkeys were enhanced by HD [43,44], and the potential risk of ARGs to golden snub-nosed monkeys was enhanced by HD [16,40], which supported the results of this study. There is an interflow of genes among livestock, wild animals, and humans, and humans can easily obtain ARGs and VFs from nonhuman primates [15]. The risk of ARGs and VFs in the habitat of golden snub-nosed monkeys was enhanced by HD, and this may be the reason why HD increased the risk of ARGs in the gut of golden snub-nosed monkeys [16]. 

MAGs containing both ARGs and VFs are defined as PARBs, and MAGs have been widely used to assess the health risk of PARBs to ecosystem environmental health [36,45,46]. Our results indicated that there were PARBs in the gut microbiota of golden snub-nosed monkeys, and the kinds and relative abundance of the PARBs were increased by HD. The subtypes and gene numbers of ARGs and VFs of the PARBs in the gut microbiota of golden snub-nosed monkeys were also increased by HD, which is supported by studies of some other animals [10,47,48,49]. The characteristics of ARGs and VFs in PARBs should be considered when the risk of PARBs in the ecological environment is evaluated. Using a factor analysis, one study found that higher contributions of VFs and ARGs in PARBs lead to a higher health risk of PARBs [35]. Our results also indicated that the health risk of PARBs to golden snub-nosed monkeys can be determined using VFs and ARGs, and the contribution of VFs to PARBs was higher than the contribution of ARGs. 

For disturbed and wild animal populations, deviations from ideal environmental conditions can be extremely harmful to health and may be associated with the occurrence and development of disease [50]. HD enhanced the health risk of PARBs to golden snub-nosed monkeys, which was mainly reflected in three aspects. First, the pathogenic mechanisms and the drug resistance scope of the PARBs were increased by HD because the biofilm, effector delivery system, exotoxin, motility of VFs, and the lincosamide, aminoglycoside, and streptogramin antibiotics of ARGs were only found in the HD group but not in the W group. Second, the risks of multidrug PARBs were increased by HD. Multidrug PARBs carrying multidrug ARGs pose a severe threat to the public health risk of superbugs [11,51], and the kinds of multidrug PARBs in the HD group were higher than those in the W group. Third, the health risks of unclassified PARBs were enhanced by HD. Unclassified PARBs of wild animals could pose unknown and serious challenges to ecosystems [47], and the kinds of unclassified PARBs in the HD group were higher than those in the W group. The three aspects indicated that the risk of bacterial infections in wild animals increased, and the scope of antibiotic resistance will grow vastly if wild animals are treated with antibiotics. More seriously, unclassified PARBs might bring unknown risks.

MGEs play an important role in the horizontal transfer of ARGs or VFs [52]. The higher the gene numbers of MGEs in a PARB, the higher the risk of the PARB becoming a superbug for humans when the PARB is transferred to human ecosystems [53]. Our results indicated that the PARBs of golden snub-nosed monkeys had the opportunity to transfer to human ecosystems, and a study by Dias et al. [14] supported our results. The risk of PARBs transfer from wild animals to human ecosystems was enhanced by HD because the total number of MGEs was enhanced by HD. ARGs or VFs can be carried in plasmids to an appropriate pathogen, which could then develop the traits of these ARGs or VFs [40]. Pathogenic ARG transfer into pathogens in human ecosystems was also increased by HD. HD enhanced the mobility rate of pathogenic ARGs in plasmids. Therefore, it is urgent to predict the transfer risk of PARBs from wild animals to humans. KEGG pathways can define the complex inter-relationships between genes and functions [54,55]. To predict the transfer risk of PARBs from wild animals to humans, the KEGG pathways of PARBs in antimicrobial drug resistance and bacterial infectious disease in humans should be studied. 

Our results indicated that HD significantly increased the ability of PARBs to cause human disease. Previous research had shown that the KEGG pathways of human disease in the gut microbiota of wild animals were enhanced with intervention for the enrichment of pathogens in the gut of wild animals, which supported our result [56]. Our result also suggested that if PARBs transfer to humans, then humans will become resistant to more classes of antibiotics. Thus, the gene numbers of MGEs and KEGG pathways numbers in antimicrobial drug resistance and bacterial infectious disease were enhanced by HD, indicating that the transfer risks of PARBs from golden snub-nosed monkeys to humans might be increased by HD. Further analysis found that if PARBs in the gut microbiota of golden snub-nosed monkeys transfer to the human ecosystem, then humans may develop additional symptoms from the pathogenic pathways of pathogenic bacteria like *Tuberculosis*, *Staphylococcus*, *Streptococcus*, *Yersinia*, *Pertussis*, and *Vibrio cholerae*. In addition, if humans are infected by these pathogens, then humans may become resistant to antifolate, platinum drugs, beta-lactam, vancomycin, and cationic antimicrobial peptides (https://www.genome.jp/kegg). There was an unclassified PARB that carried multidrug ARGs in the gut of golden snub-nosed monkeys that was G_bin134, which indicated that the transfer of the unclassified PARBs from wild animals also may cause unknown diseases in humans. Therefore, it is necessary to monitor PARBs from wild animals and develop emergency measures to treat human diseases. 

However, due to the common limitation of incomplete MAGs [28], no MAGs were identified at the species level in our study. The transfer of PARBs from wild animals to humans is related to the species of PARBs in wild animals, the characteristics of habitat environments, and the number of humans exposed to the PARBs. Thus, in the future, the amount and sequencing depth of fecal samples of wild animals should be increased to identify the species of PARBs and the pathogenicity and drug resistance of the PARBs. What is more, the transfer path of PARBs from wild animals to humans and the collinearity of the PARBs and the microbiome of human hosts need to be studied under the eco–environment coupling system of wild animals and humans. 

## 5. Conclusions

HD significantly affected the risk of ARGs to the gut microbiota of golden snub-nosed monkeys. The total relative abundance of PARBs and the subtypes and gene numbers of ARGs, VFs, and MGEs were increased by HD. Golden snub-nosed monkeys can be used as sentinel wild animals to investigate the health risk of HD to wild animals and the transfer risk of PARBs from wild animals to humans. On the one hand, regarding health risks to golden snub-nosed monkeys, the contribution of ARGs and VFs of PARBs was enhanced by HD. On the other hand, the gene numbers of KEGG pathways in antimicrobial drug resistance and bacterial infectious disease of PARBs were increased by HD. The health risk to golden snub-nosed monkeys was mainly reflected in the increase in the species and abundance of PARBs, the emergence of multiple PARBs, and the unknown risk of unclassified PARBs. The transfer risk of PARBs to humans was mainly reflected in newly added pathogenic pathways of PARBs to humans. We recommend reducing the conflicts between wild animals and humans. Moreover, the characteristics of ARGs, VFs, and KEGG pathways of PARBs in wild animals need to be investigated under the eco–environment coupling system of wild animals and humans, which could be used to predict the risks of PARBs to wild animals and find the proper antibiotics to treat wild animal and human diseases caused by PARBs.

## Figures and Tables

**Figure 1 animals-13-03083-f001:**
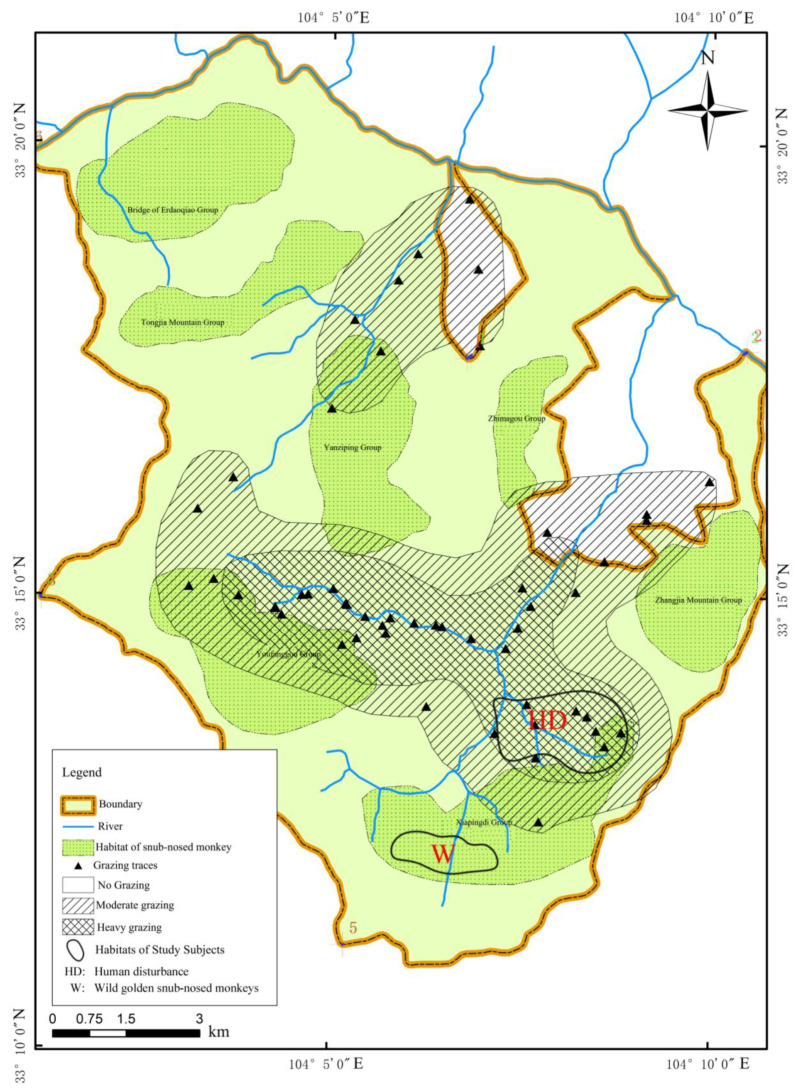
Baihe National Nature Reserve and sampling points.

**Figure 2 animals-13-03083-f002:**
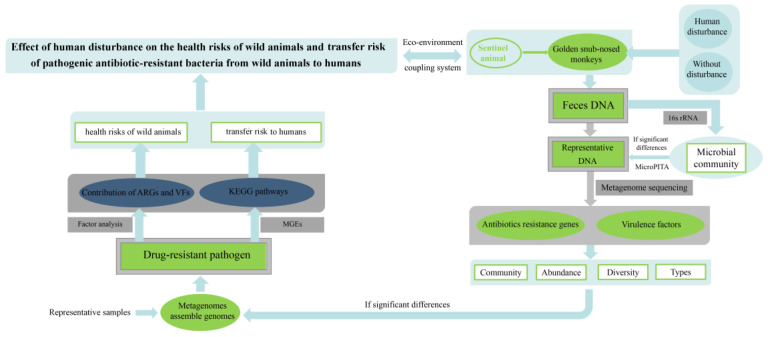
Conceptual framework of a health risk assessment.

**Figure 3 animals-13-03083-f003:**
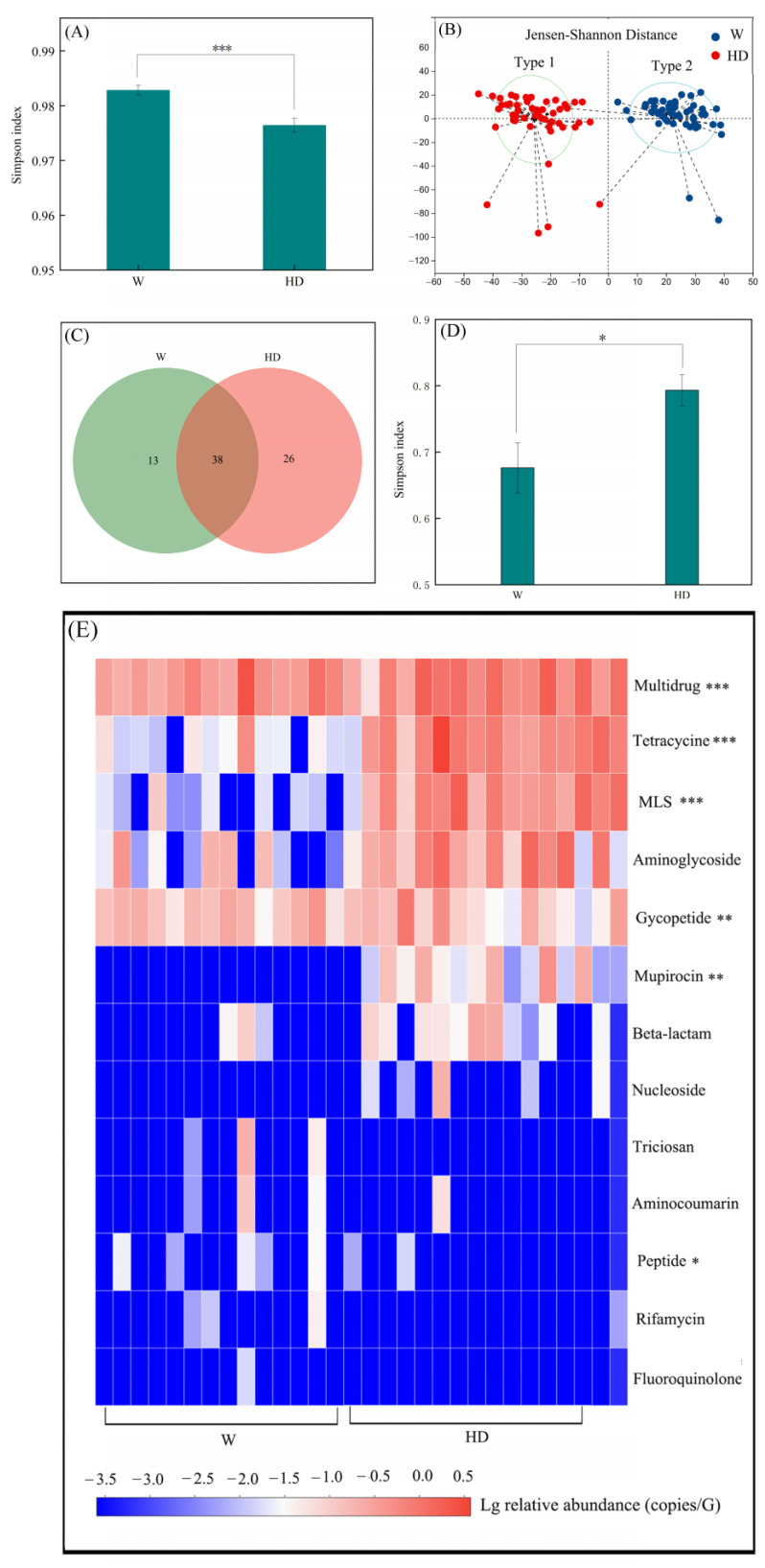
The effect of HD on ARGs in the gut microbiota of golden snub-nosed monkeys. (**A**,**B**) Simpson diversity indices functional types of the gut microbiota of golden snub-nosed monkeys. (**C**) The subtypes of ARGs. (**D**) Simpson diversity indices of ARGs. (**E**) The relative abundance of ARG classes. The *, **, and *** indicate significant differences between the HD group and the W group at the 0.05, 0.01, and 0.001 levels, respectively.

**Figure 4 animals-13-03083-f004:**
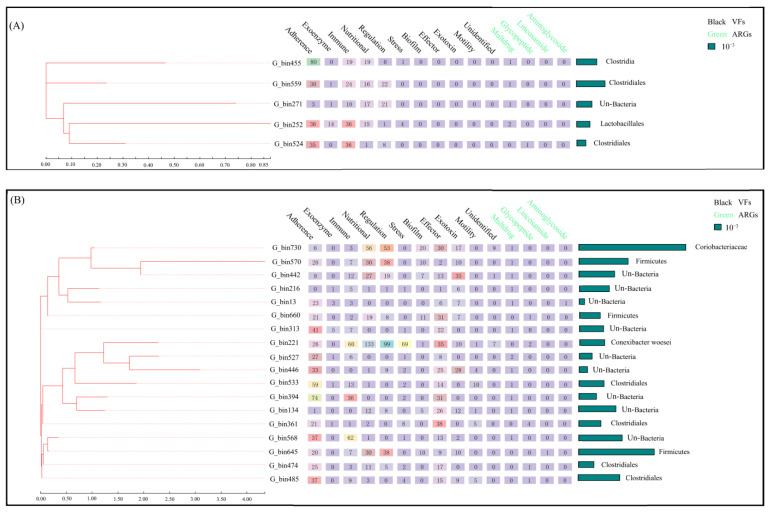
Characteristics and relative abundance of PARBs. (**A**,**B**) The characteristics and relative abundance of PARBs in the W and HD groups.

**Figure 5 animals-13-03083-f005:**
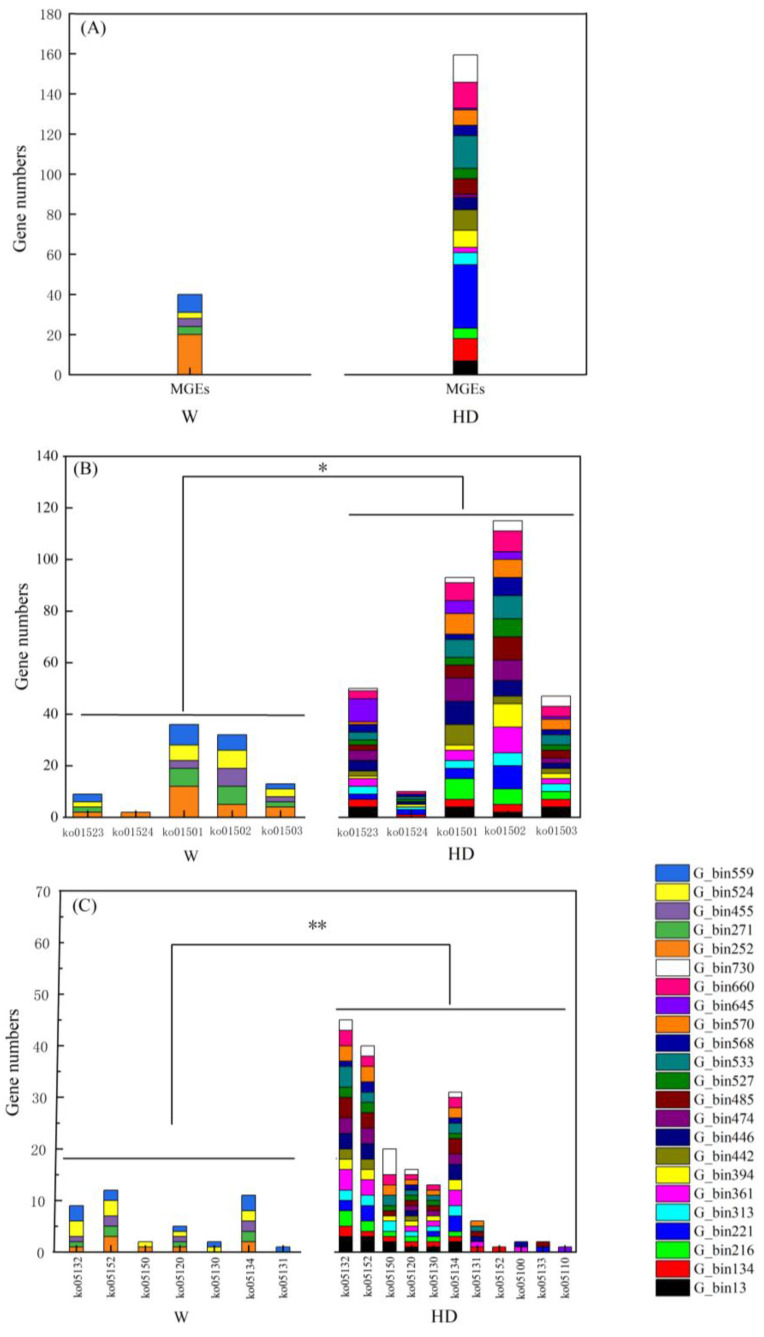
The gene numbers of MGEs, KEGG pathways of antimicrobial drug resistance, and bacterial infectious disease in PARBs. (**A**) Gene numbers of MGEs in PARBs in the HD group and W group. (**B**,**C**) Gene numbers of KEGG pathways in bacterial infectious disease antimicrobial drug resistance in the HD group and the W group, respectively. The * and ** indicate significant differences between the HD group and the W group at the 0.05 and 0.01 levels, respectively.

**Table 1 animals-13-03083-t001:** The contigs numbers and mobility of pathogenic ARGs.

Contig	Antibiotic Resistance Genes (ARGs)	ARG Class	Virulence Factors (VFs)	Mobile Genetic Element Genes (MGEs)	Contig Numbers in Wild Individuals (W Group)	Contig Numbers in Human Disturbance Individuals (HD Group)
HD50_k97_208870_1	acrB	Multidrug	AcrAB	Plasmid	6	0
HD7_k97_107630_1	ceoB	Multidrug	AdeFGH	Plasmid	54	2
HD7_k97_185188_1	adeF	Multidrug	AdeFGH	Plasmid	2	0
HD7_k97_208861_1	ceoB	Multidrug	AdeFGH	Plasmid	6	0
HD7_k97_226438_1	ceoB	Multidrug	AdeFGH	Plasmid	10	2
HD7_k97_309577_1	ceoB	Multidrug	AdeFGH	Plasmid	92	0
HD7_k97_315010_1	adeF	Multidrug	AdeFGH	Plasmid	18	0
HD7_k97_334322_1	adeF	Multidrug	AdeFGH	Plasmid	18	80
HD7_k97_375936_1	adeF	Multidrug	AdeFGH	Plasmid	6	0
HD7_k97_401148_1	adeF	Multidrug	AdeFGH	Plasmid	24	0
W34_k97_1150608_1 ^#^	MexB	Multidrug	AcrAB	Plasmid	0	12
W34_k97_132510_1	adeF	Multidrug	AdeFGH	Plasmid	0	12
W34_k97_327771_1	acrB	Multidrug	AcrAB	Plasmid	4	58
W34_k97_36958_1	acrB	Multidrug	AcrAB	Plasmid	0	4
W34_k97_551122_1 ^#^	adeF	Multidrug	AdeFGH	Plasmid	0	6
W34_k97_58465_1	ceoB	Multidrug	AdeFGH	Plasmid	0	12
W34_k97_590319_1	adeF	Multidrug	AdeFGH	Plasmid	2	12
W34_k97_713320_1	adeF	Multidrug	AdeFGH	Plasmid	0	24
W34_k97_727054_1	ceoB	Multidrug	AdeFGH	Plasmid	0	20
W34_k97_939152_1 ^#^	adeF	Multidrug	AdeFGH	Plasmid	0	6
W42_k97_49042_1 ^#^	ugd	Peptide	Capsule	Plasmid	0	6
W42_k97_62472_1	smeE	Multidrug	AcrAB	Plasmid	2	24
W44_k97_574071_1	acrB	Multidrug	AcrAB	Plasmid	2	18
W7_k97_500225_1	ugd	Peptide	Capsule	Plasmid	0	16
Mobility Rate (%)					58.33	78.33

Note: ^#^ The MGEs in drug resistance genes had integrative conjugative elements.

**Table 2 animals-13-03083-t002:** Risk of infectious diseases from PARBs.

Groups		F_1_	F_2_	F_3_	F_4_	F_5_	F_6_	TF	HR (10^−3^)	THR (10^−3^)
W	G_bin455	−0.267	−0.667	0.026	−0.180	−1.940	0.078	−0.3746	−11.10	−28.5
	G_bin559	0.013	−0.793	−0.480	−0.082	−0.370	0.081	−0.2138	−8.90	
	G_bin271	−0.127	−0.657	−0.602	−0.029	0.727	0.008	−0.1319	−3.00	
	G_bin252	0.247	−1.437	−2.218	2.150	0.862	1.010	−0.0489	−0.960	
	G_bin524	−0.120	−0.135	−0.900	−0.989	−0.577	−0.125	−0.3362	−4.45	
HD	G_bin730	1.333	−0.191	2.278	0.829	1.35	0.811	0.8834	138.6	125.8
	G_bin570	0.120	−0.787	0.652	−0.148	0.417	0.257	0.0455	3.280	
	G_bin442	0.081	−0.299	1.700	0.455	0.491	−0.085	0.288	15.05	
	G_bin216	−0.500	−0.448	−0.346	0.140	0.880	−0.079	−0.1362	−5.96	
	G_bin13	−0.331	−0.569	−0.002	0.290	0.394	−4.322	−0.3845	−3.02	
	G_bin660	−0.293	0.404	0.756	−0.547	0.559	0.336	0.1038	3.14	
	G_bin313	−0.417	−0.068	−0.636	0.661	0.171	0.432	−0.0822	−2.90	
	G_bin221	4.143	0.933	−0.579	−0.197	−0.626	−0.462	0.9991	36.7	
	G_bin527	−0.504	−0.584	−0.213	0.877	0.071	0.276	−0.1224	−2.29	
	G_bin446	−0.620	0.856	1.535	0.672	−0.399	−0.085	0.1856	2.25	
	G_bin533	−0.850	1.192	0.399	0.578	−1.56	0.485	−0.0619	−2.27	
	G_bin394	−0.171	0.050	0.221	0.150	−2.04	0.321	−0.1812	−4.67	
	G_bin134	−0.413	0.301	0.685	0.355	0.947	0.191	0.1735	9.48	
	G_bin316	−0.443	3.035	−1.477	−0.347	1.39	0.181	0.2539	8.21	
	G_bin568	0.106	−0.620	−0.312	0.015	−1.464	0.184	−0.2394	−15.24	
	G_bin645	−0.031	−1.390	0.078	−3.46	0.729	0.702	−0.4365	−48.4	
	G_bin474	−0.358	0.598	−0.613	−0.757	0.397	−0.118	−0.1294	−2.91	
	G_bin485	−0.598	1.277	0.047	−0.436	−0.399	−0.074	−0.0536	−3.27	

Note: F_1_, F_2_, F_3_, F_4_, F_5_, F_6_, TF, SR, and TSR are the scores of principal component one, principal component two, principal component three, principal component four, principal component five, principal component six, total score of all principal components for the health risk of each MAG, and total health risk of PARBs to the HD group and the W group, respectively. The “−” in the table does not represent a value, it represents the magnitude of the direction.

## Data Availability

The datasets presented in this study can be found in online repositories. The names of the repository/repositories and accession number(s) can be found at: https://submit.ncbi.nlm.nih.gov/subs/sra/SUB12982665/overview, PRJNA947971 and https://submit.ncbi.nlm.nih.gov/subs/sra/SUB12971514/overview, PRJNA947945.

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
