# Peer review of "Human Disturbance Increases Health Risks to Golden Snub-Nosed Monkeys and the Transfer Risk of Pathogenic Antibiotic-Resistant Bacteria from Golden Snub-Nosed Monkeys to Humans"

_animals, 2023, doi:10.3390/ani13193083_

Round 1

Reviewer 1 Report

This manuscript presents a relevant study regarding the use of golden snub-nosed monkeys (Rhinopithecus roxellana) as a sentinel of antibiotic resistance and the effects of human disturbance in this resistance pattern. I believe this is an emergent and crucial topic to be addressed nowadays and authors should receive credit for this. In general, I believe the work is confusing, difficult to follow and to realize what the true goals were because I can see that there were at first some experiments and then you have conducted your main research project. But this is hard to follow this way.  

Here I present some point-by-point corrections to improve the manuscript.

1) Authors should provide a Simple Summary of their work, like a short comprehensive abstract that helps people from other research areas to address it.

2) The structure of the abstract is confusing for me. Authors talk about their aims/goals in distinct parts of it, providing no short information on the methods used. It is not correctly organised in background, methods, results, conclusions (without the subheadings).

3) This sentence is too simple/basic and it says absolutely nothing in the way it is written: "Wild animals are included in “One Health”. (L14). Please Rephrase in a way that you can include your research in the One Health approach.

4) L29-30 I don't understand the nomenclature or italic typing of these

5) I do not completely agree with the definition of sentinel species that authors use in their work. I believe they should check the concept of "bioindicator" and then try to understand which concept fits better and use it for golden monkeys. In the same way, I do not understand why authors believe the golden monkeys are good for this work. Authors should present positive criteria for choosing this species (easy to sample, manipulate, easy to sample, good and visible changes as a consequence of the studied variable....), they are described in the literature. Being a flag species and regulated are not the most correct scientific arguments.

6) Once again I believe the last paragraph of the introduction needs to be completely rephrased. Authors can try to number their goals to explain them a little better. Why do you start it with "In conclusion"? It is not a good connector to use here. In this paragraph,  authors provide methods and conclusions and this information should be part of other article sections. For me it was really difficult to follow.

Author Response

Reviewer #1:

Comment (1): Authors should provide a Simple Summary of their work, like a short comprehensive abstract that helps people from other research areas to address it.

Response to comment (1): I apologize for our carelessness. See lines 14-27, the Simple Summary section has been added before the Abstract chapter. Thank you for your valuable comments.

Comment (2): The structure of the abstract is confusing for me. Authors talk about their aims/goals in distinct parts of it, providing no short information on the methods used. It is not correctly organized in background, methods, results, conclusions (without the subheadings).

Response to comment (2): Thank you for your kind reminder, we are so sorry for the confusing structure of the abstract. In order to make the abstract more clearly, the abstract section has been rephrased, and the words of "Background", "Aim", "Study design and methods", "Results", and "Conclusion" have been added in suitable place of the abstract section. We apologize for our carelessness.

Comment (3): This sentence is too simple/basic and it says absolutely nothing in the way it is written: "Wild animals are included in "One Health". (L14). Please Rephrase in a way that you can include your research in the One Health approach.

Response to comment (3): We apologize for our inappropriate expression, the sentence that "Wild animals are included in "One Health"" is part of the sentence that "The control and research of pathogenic microorganisms that can cause disease in animals and humans is the core content of "One Health". In order to better reflect the importance of wild animals and their pathogenic microorganisms in the "One Health", we have revised the above sentences to that "From the perspective of interactions in human-animal-ecosystem, the control and research of pathogenic bacteria that can cause disease in animals and humans is the core content of "One Health"" in lines 29-31. We hope that our modifications meet your requirements, but if there is anything inappropriate, please feel free to let us know.

Comment (4): L29-30 I don't understand the nomenclature or italic typing of these

Response to comment (4): We apologize for our unclear expression. What we want to express is that these pathogens may have pathogenic effects on wild animals through biofilm, effector delivery system, exotoxin, motility of wild animals and HD enhanced the pathogenic effects. In order to express this sentence more clearly, the sentence that "HD increased the pathogenicity of PARBs to wild animals in terms of biofilm, effector delivery system, exotoxin, motility and resistance to lincosamide, aminoglycoside, and streptogramin antibiotics.” have be modified to that "HD increased the pathogenicity of PARBs to wild animals and the PARBs in wild animals will resistance to lincosamide, aminoglycoside, and streptogramin antibiotics.". See lines 47-49.

    Italic typing in the lines51 to 52, and they are pathogenic bacteria. We are sorry our mistake, the italic typing of the pathogenic bacteria has been altered, and the sentence of "If these PARBs spill over to humans, new KEGG pathways of bacterial infectious disease were induced by HD from the PARBs, including tubercle bacillus, Staphylococcus, Streptococcus, Yersinia, pertussis, and Vibrio cholera" has been changed to the sentence of "If these PARBs spilled over from the golden snub-nosed monkeys to humans, the humans may be acquired symptoms of the pathogens of tubercle bacillus, Staphylococcus, Streptococcus, Yersinia, pertussis, and Vibrio cholera.". See lines 49-52.

We ensure that our revisions do not change the theme and meaning of the article. We hope that our modifications meet your requirements, but if there is anything inappropriate, please feel free to let us know.

Comment (5): I do not completely agree with the definition of sentinel species that authors use in their work. I believe they should check the concept of "bioindicator" and then try to understand which concept fits better and use it for golden monkeys. In the same way, I do not understand why authors believe the golden monkeys are good for this work. Authors should present positive criteria for choosing this species (easy to sample, manipulate, easy to sample, good and visible changes as a consequence of the studied variable....), they are described in the literature. Being a flag species and regulated are not the most correct scientific arguments.

Response to comment (5): We apologize for our inappropriate expression, and thank you for your valuable comment. We have carefully studied the concept of "bioindicator" and "sentinel animals".

Bioindicator refers to those organisms that can indicate the environment or the characteristics of a certain environmental factor through changes in their characteristics, quantity, species or community within a certain area. In environmental protection, some sensitive bioindicators are commonly used to indicate environmental pollution. Bioindicators generally refer to specific species sensitive to environmental changes, include indicator plants, plankton, periphyton, benthic animals, fish and bacteria and so on[1].

Sentinel animal is a kind of animal used to monitor or warn the pollution degree of toxic and harmful substances or potential toxic and harmful substances in a specific area[2]. In previous published studies, the animals, such as wild rat[3], dog[4], wild boar[5-7], birds[8], bison population[9], Chile[10] and so on, could be used as sentinel animal to assess the risk of pathogenic bacteria and their ARGs and study the impact of HD on the environmental health under "One Health". The purpose of this manuscript is to study the impact of HD on the health risks of wild animals and the risk of spillover of pathogenic bacteria from wild animals to humans, which is the core content of "One Health". Thus, we used the golden snub-nosed monkeys as sentinel animals in this study and we thought that the sentinel animals could be better reflect the purposes of this manuscript.

    However, we fully accept your comment that we should present positive criteria for choosing this species (golden snub-nosed monkeys). Therefore, we carefully reviewed the literature on the intestinal microbes in golden snub-nosed monkeys[11-15], and we added the sentence that "In order to conduct research on golden snub-nosed monkeys (Rhinopithecus roxellana), some of them have been fixed in a place of their wild habitat, so their feces are easy to sample and manipulate. As well as they are highly regulated by humans, thus there are good and visible changes in intestinal microbes as a consequence of the studied variable" in lines 89-93 to described why the golden snub-nosed monkeys were choose as sentinel animals in revised manuscript. The sentence that "Golden snub-nosed monkeys (Rhinopithecus roxellana) are the flagship species of the world natural heritage site in China, and they are highly regulated by humans" has been deleted.

We appreciate your kind reminder very much, and we ensure that our revisions do not change the theme and meaning of the article. We hope that our modifications meet your requirements, but if there is anything inappropriate, please feel free to let us know.

Comment (6): Once again I believe the last paragraph of the introduction needs to be completely rephrased. Authors can try to number their goals to explain them a little better. Why do you start it with "In conclusion"? It is not a good connector to use here. In this paragraph, authors provide methods and conclusions and this information should be part of other article sections. For me it was really difficult to follow.

Response to comment (6): We appreciate your kind comment, and we fully accept this comment. We have carefully studied other articles published in the journal of "Animals", last paragraph of the introduction section be modified to the following sentences: In this study, the characteristics of PARBs in gut microbiota of golden snub-nosed monkeys were analyzed by metagenomic sequencing, which was used to study the effect of HD on health risks of PARBs to wild animals and the spillover risk of PARBs from wild animals to humans. Our results showed that HD increased the pathogenicity of PARBs to wild animals and the PARBs in wild animals will resistance to lincosamide, aminoglycoside, and streptogramin antibiotics. If these PARBs spill over from the golden snub-nosed monkeys to humans, the humans may be acquired symptoms of the pathogens of Tubercle bacillus, Staphylococcus, Streptococcus, Yersinia, pertussis, and Vibrio cholera which could guide to develop the management in the treatment of diseases in wild animals and emergency measures for spillover of pathogens to humans. See lines 103-113.

We hope that our modifications meet your requirements, but if there is anything inappropriate, please feel free to let us know.

Reviewer 2 Report

Please, find below the clarification to be addressed and some suggestions for your interesting article.

1. the ARGs generatedcaused by HD may be transferred to wild animals (Campbell et al., 57 2020)

2. Provide reference for sentence 67-68. Mobile genetic elements (MGEs) can carry ARGs or VFs of pathogenic bacteria to that enter the eco-environment.

3. Provide reference and revise sentence 68 - 70. An antibiotic-resistant bacteria do not increase human resistance to antibiotics but can cause the emergence and/or spread of antibiotic-resistant bacteria infections in humans. Consult:https://www.who.int/news-room/fact-sheets/detail/antibiotic-resistance.

If pathogenic antibiotic-resistant bacteria (PARBs) in wild animals spill over to humans, they can cause new diseases in the human ecosystem or increase human resistance to antibiotics. 

4. Define clearly in the paragraph the wild habitat area. Is the golden snub-nosed monkey different from ground monkey.

Revise 113-114: Xiaping ground monkeys group live around Taiping village that is is a gathering place for indigenous people in the reserve, Xiaping ground monkey group has been severely disturbed by grazing (Figure 1).

Thus, in this study, the Xiapingdi monkey groups were the study subjects (Figure 1). 

5. Clarify the highlighted text: According to the results of the 16S rRNA sequencing, the DNA of 15 and 15 representative samples was screened from the W and HD groups for meta- 158 genomic sequencing analysis (Table S1)

6. Specify in text the Ri of Mi. It is not clear.

7. Clarify the formula 2, 3, 4, 5 in the lines from 219 to 222, and clarify in text: Fi, Ri, THRH and Ri.

8. Provide figures in text. (249) 

The Simpson diversity indices of gut microbes in the HD group were significantly lower than those in the W group (Figure 2A; P <0.001).

9. Provide figures in the highlighted text.

The gene numbers of MGEs were 40 and 186 in the PARBs of the W group and HD group, respectively (Figure 4A), and the total gene numbers of KEGG pathways in bacterial infectious disease and antimicrobial drug resistance in the HD group were higher than those in the W group (P<0.05).

10. Clarify the highlighted text. 

The KEGG pathways that were only checked in the PARBs in the HD group but not in the W group involved KEGG pathways in bacterial infectious disease of ko05152, ko05100, ko05133 and ko05110, and their host PARBs could produce antimicrobial drug resistance to antibiotics by KEGG pathways of ko01523, ko01524, ko01501, ko01502 and ko01503.

11. Clarify the highlighted text. 

Second, the risks of multidrug PARBs were increased by HD. Multidrug PARBs carry multidrug resistance genes and are a severe threat to public health (Kim et al., 2021), and multidrug PARBs increase the risk of superbugs to humans (Dunn et al., 2019). In the HD group, 12 MAGs were multidrug resistant PARPs, while 4 MAGs were multidrug resistant PARBs in the W group.

There were 8 species of PARBs that were unclassified, and all of them were multidrug PARBs in the gut of the HD group, while there was only one species of PARB in the gut of the W group.

12. Because the ability of pathogenic ARGs to transfer horizontally to PARBs by plasmids and the numbers of MGEs in PARBs were increased by HD, the opportunity possibility for PARBs to spill into human ecosystems was increased by HD. 

13. Please, clarify that you evaluated the risk in monkeys and deduced the result for human in lines 422 to 433.

14. Simplify the sentence:

The KEGG pathways that were only assessed in the PARBs in the HD group but not in the W group involved the risk of infection by tuberculosis, Staphylococcus, Streptococcus, Yersinia, pertussis, and Vibrio cholerae in golden monkey populations was enhanced by HD, and if human beings were infected by those pathogenic bacteria, the pathogenic bacteria may be resistant to antifolate, platinum drugs, beta-lactam, vancomycin, and cationic antimicrobial peptide (https://www.genome.jp/kegg).

15. Clarify the highlighted text. 

Fortunately, the bacterial infectious disease and antimicrobial drug resistance to antibiotics of the PARBs in the gut of wild animals could be predicted by MAG technology through the KEGG system of humans.

16. Is that possible to mention the measures to be addressed?

The results confirmed our hypothesis, and the next steps are to formulate measures for health management and rescue of wild animals and obtain address emergency measures to prevent spillover of drug-resistant pathogens from wild animals.

17. Is that possible to provide a Conceptual Framework of the health risk assessment?

See previous box

Author Response

Dear Editors and Reviewers:

Thank you for your letter and for the comments concerning our manuscript entitled "Human disturbance increases the health risks of golden snub-nosed monkeys and the spillover risks of pathogenic antibiotic-resistant bacteria from golden snub-nosed monkeys to humans" (ID: animals-2533952). Those comments were all valuable and very helpful for revising and improving our paper and provided important guiding significance for our research. We have studied the comments carefully and made corrections that we hope meet with approval. The revised portions are marked in red in the paper. The main corrections in the paper and the responses to the reviewer’s comments are as follows:

Reviewer #2:

Comment (1): the ARGs generated caused by HD may be transferred to wild animals (Campbell et al., 57 2020).

Response to comment (1): We apologize for our unclear expression, in order to express this meaning more clearly, we modified this sentence that "the ARGs generated caused by HD may be transferred to wild animals" to "the ARGs may be transferred to wild animals from the human disturbed habitat of wild animals". See lines 73 and 74.

Comment (2): Provide reference for sentence 67-68. Mobile genetic elements (MGEs) can carry ARGs or VFs of pathogenic bacteria to that enter the eco-environment.

Response to comment (2): Mobile genetic elements (MGEs) can carry ARGs or VFs of pathogenic bacteria to enter the eco-environment, which had been researched by other authors and published in their references, and we 're sorry we didn 't cite the references. We thank you for your kind reminder, and we have added the suitable references for this sentence[16].

Comment (3): Provide reference and revise sentence 68-70.

Response to comment (3): We are so sorry that we didn 't cite the references in this place. We thank you for your kind reminder, the references [17] has been cited in the revised manuscript. As well as, the sentence that "If pathogenic antibiotic-resistant bacteria (PARBs) in wild animals spill over to humans, they can cause new diseases in the human ecosystem or increase human resistance to antibiotics." has been changed to that "If the PARBs in wild animals spill over to humans, they may infect the humans and cause emergencies" See lines 81 and 82. We apologize for our incorrect expression.

Comment (4): Define clearly in the paragraph the wild habitat area. Is the golden snub-nosed monkey different from ground monkey.

Revise 113-114: Xiaping ground monkeys group live around Taiping village that is a gathering place for indigenous people in the reserve, Xiaping ground monkey group has been severely disturbed by grazing (Figure 1). Thus, in this study, the Xiapingdi monkey groups were the study subjects (Figure 1).

Response to comment (4): In our study, the golden snub-nosed monkeys in HD group and W group are the parts of Xiapingdi golden snub-nosed monkeys, but the habitats of HD group is different from that of W group.

We apologize for our confused expression. Thank you for your valuable comments that let us realize that we did not explain clearly why we chose the Xiapingdi golden snub-nosed monkeys in the Baihe National Nature Reserve. In order to improve the quality of this manuscript, the sentence that

"Xiaping ground monkeys group live around Taiping village that is a gathering place for indigenous people in the reserve, Xiaping ground monkey group has been severely disturbed by grazing (Figure 1). Thus, in this study, the Xiapingdi monkey groups were the study subjects (Figure 1)." has been changed to that " The Baihe National Nature Reserve has the highest density, largest and most representative population of golden snub-nosed monkeys so far. The home ranges of golden snub-nosed monkeys in Baihe National Nature Reserve were marked by the light green in Figure 1. The fourth national survey of giant pandas found that the situation of grazing in the reserve was serious, and HD trace points, especially grazing trace points overlapped with the home range of golden snub-nosed monkeys. The golden snub-nosed monkeys in some families in the home range of Xiapingdi have been severely disturbed by grazing, and they were fixed in a place of their home range in 2021 for ecotourism and scientific research, thus they have been disturbed by humans (HD group). The other golden snub-nosed monkeys in the home range of Xiapingdi are living in mountains, the mountains are inaccessible and they never are disturbed by humans (W group). There were HD group and W group in the home range of Xiapingdi, so the HD group and W group of Xiapingdi golden snub-nosed monkeys were choose as the study subjects in this study. ". See lines 117-130.

We appreciate your kind reminder very much, and we ensure that our revisions do not change the theme and meaning of the article. We hope that our modifications meet your requirements, but if there is anything inappropriate, please feel free to let us know.

Comment (5): Clarify the highlighted text: According to the results of the 16S rRNA sequencing, the DNA of 15 and 15 representative samples was screened from the W and HD groups for metagenomic sequencing analysis (Table S1)

Response to comment (5): Thank you for your kind reminder, we apologize for our confused expression in this sentence. In order to more clearly express the meaning of this sentence, the sentences that "To ensure that metagenomic sequencing technology can fully obtain all the information on the gut microbiome of Sichuan golden snub-nosed monkeys, the selection criterion of "most representative" in the micro-PITA method was used before metagenomic sequencing. According to the results of the 16S rRNA sequencing, the DNA of 15 and 15 representative samples was screened from the W and HD groups for metagenomic sequencing analysis (Table S1). " have been changed to that " Representative samples were selected according to β diversity of gut microbiota to fully obtain all information on the gut microbiota of golden snub-nosed monkeys. According to the results of the 16S rRNA sequencing, 15 and 15 representative DNA samples of gut microbiota were selected from the 50 and 53 DNA samples for metagenomic sequencing analysis in the W and HD groups, respectively, by the selection criterion of "most representative" in micro-PITA method, and the representative samples were showed in Table S1. " See lines 164-170. We hope that our modifications meet your requirements, but if there is anything inappropriate, please feel free to let us know.

Comment (6): Specify in text the Ri of Mi. It is not clear.

Response to comment (6): We apologize for our confused expression in this section. Ri (the "Ri" has been changed to "Ai" in the revised manuscript) represents the relative abundances of i MAG, and there is no "Mi" in formula (1), thus the "Mi" has been deleted in revised manuscript. Thank you for your kind reminder.

Comment (7): Clarify the formula 2, 3, 4, 5 in the lines from 219 to 222, and clarify in text: Fi, Ri, THRH and Ri.

Response to comment (7): There are 4 formulas in this manuscript, we apologize for our mistake in this section. The correct formula numbers have been modified, and the formula numbers in other places has also been modified in revised manuscript.

We apologize for the ambiguity caused by abbreviations in the formulas, such as the Ri, HRi and THR, as well as Fmi and Wmi. After our careful discussion, the ambiguous abbreviations in the formulas have been changed, the Ri has been changed to Ai, the Fmi has been changed to Si, Wmi has been changed to Wi. See lines 240-246. And the abbreviations of formula have also been modified in other places of revised manuscript. We appreciate your kind reminder very much, and we ensure that our revisions do not change the meaning of formulas. We hope that our modifications meet your requirements, but if there is anything inappropriate, please feel free to let us know.

Comment (8): Provide figures in text.

Response to comment (8): Thank you for your kind reminder, the figures have been provided in revised manuscript. And the sentence that "The Simpson diversity indices of gut microbes in the HD group were significantly lower than those in the W group (Figure 2A; P <0.001)." has been changed to that "Figure 3A showed that the Simpson diversity indices of gut microbes in the HD group were significantly lower than those in the W group (P <0.001).". See lines 272 AND 272. We are so sorry for our mistake in this section.

Comment (9): Provide figures in the highlighted text.

Response to comment (9): Thank you for your kind reminder, the figures have been provided in revised manuscript.

And the sentence that "The gene numbers of MGEs were 40 and 186 in the PARBs of the W group and HD group, respectively (Figure 4A), and the total gene numbers of KEGG pathways in bacterial infectious disease and antimicrobial drug resistance in the HD group were higher than those in the W group (P<0.05)." has been changed to that " Figure 5A showed that the total gene numbers of MGEs were 40 and 186 in the PARBs of the W group and HD group, respectively. Figure 5B and Figure 5C showed that the total gene numbers of KEGG pathways in bacterial infectious disease and antimicrobial drug resistance in the HD group were higher than those in the W group (P<0.05; P<0.01). ". See lines 343-347. We hope that our modifications meet your requirements, but if there is anything inappropriate, please feel free to let us know.

Comment (10): Clarify the highlighted text.

Response to comment (10): We apologize for our confused expression. The highlighted text that "The KEGG pathways that were only checked in the PARBs in the HD group but not in the W group involved KEGG pathways in bacterial infectious disease of ko05152, ko05100, ko05133 and ko05110, and their host PARBs could produce antimicrobial drug resistance to antibiotics by KEGG pathways of ko01523, ko01524, ko01501, ko01502 and ko01503." has been changed to that "The KEGG pathways of bacterial infectious disease that were only checked in their host PARBs in the HD group were ko05152, ko05100, ko05133 and ko05110, their host PARBs could produce antimicrobial drug resistance to antibiotics by KEGG pathways of ko01523, ko01524, ko01501, ko01502 and ko01503.". See 347-351. We are so sorry for our mistake in this section, and thank you very much for your kind reminder.

Comment (11): Clarify the highlighted text.

Response to comment (11): We apologize for our confused expression. The highlighted text that "Second, the risks of multidrug PARBs were increased by HD. Multidrug PARBs carry multidrug resistance genes and are a severe threat to public health, and multidrug PARBs increase the risk of superbugs to humans. In the HD group, 12 MAGs were multidrug resistant PARPs, while 4 MAGs were multidrug resistant PARBs in the W group." has been changed to that " Second, the risks of multidrug PARBs were increased by HD. Because the multidrug PARBs carry multidrug ARGs and they have a severe threat to public health for the risk of superbugs, and the kinds of multidrug PARBs of the HD group were higher than that of W group.in lines 402-406. And the sentence that "In the HD group, 12 MAGs were multidrug resistant PARPs, while 4 MAGs were multidrug resistant PARBs in the W group." was the description of Figure 3, and it was placed in the 3.2 section, see lines 303-305.

The highlighted text that "There were 8 species of PARBs that were unclassified, and all of them were multidrug PARBs in the gut of the HD group, while there was only one species of PARB in the gut of the W group." has been changed to that "There were 8 species of PARBs of in the gut of HD group were unclassified, while that was only one species in the gut of the W group." . We also found that the sentence is a description of Figure 3 and should be placed in the 3.2 section. See 302 and 303. However, the sentence that " and the kinds of unclassified PARBs of the HD group were higher than that of the W group." has been used to replace the highlighted text in the 4 section. See lines 405 and 406.

We fully accept your comments, and thank you very much for your kind reminder. But if there is anything inappropriate, please feel free to let us know.

Comment (12): Because the ability of pathogenic ARGs to transfer horizontally to PARBs by plasmids and the numbers of MGEs in PARBs were increased by HD, the opportunity possibility for PARBs to spill into human ecosystems was increased by HD.  

Response to comment (12): We apologize for our confused expression. The sentence that "Because the ability of pathogenic ARGs to transfer horizontally to PARBs by plasmids and the numbers of MGEs in PARBs were increased by HD, the opportunity possibility for PARBs to spill into human ecosystems was increased by HD." has been changed to that "The opportunity for pathogenic ARGs to spill into pathogen in human ecosystems was also increased by HD, for HD enhanced the mobility rate of pathogenic ARGs by the plasmids". Thank you very much for your kind reminder. See lines 422-425.

Thank you very much for your kind reminder. But if there is anything inappropriate, please feel free to let us know.

Comment (13): Please, clarify that you evaluated the risk in monkeys and deduced the result for human in lines 422 to 433.

Response to comment (13): The risk in golden snub-nosed monkeys and humans is due to changes in the characteristics of PARBs in gut of golden snub-nosed monkeys in our study. We apologize for that the relationships between the health risks of golden snub-nosed monkeys and human did not be showed in this section. So the sentence that has been changed to that " If the PARBs in gut microbiota of golden snub-nosed monkeys spill into the human ecosystem, humans may get additional symptoms from the pathogenic pathways of pathogenic bacteria that Tuberculosis, Staphylococcus, Streptococcus, Yersinia, pertussis, and Vibrio cholerae, and if humans are infected by pathogenic pathways, humans may resistant to antifolate, platinum drugs, beta-lactam, vancomycin, and cationic antimicrobial peptide (https://www.genome.jp/kegg). There was an unclassified PARB carried multidrug ARGs in gut of golden snub-nosed monkeys that was the G_bin134, which indicated that the spillover of the unclassified PARBs from wild animals also may cause unknown diseases to humans. Therefore, it is necessary to monitor PARBs from wild animals and develop emergency measures to treat human disease. " in lines 434-444.

We ensure that our revisions do not change the theme, contents aims and result of the article. We hope that our modifications meet your requirements, but if there is anything inappropriate, please feel free to let us know.

Comment (14): Simplify the sentence:

Response to comment (14): We apologize for our confused expression. The highlighted text that "The KEGG pathways that were only assessed in the PARBs in the HD group but not in the W group involved the risk of infection by tuberculosis, Staphylococcus, Streptococcus, Yersinia, pertussis, and Vibrio cholerae in golden monkey populations was enhanced by HD, and if human beings were infected by those pathogenic bacteria, the pathogenic bacteria may be resistant to antifolate, platinum drugs, beta-lactam, vancomycin, and cationic antimicrobial peptide (https://www.genome.jp/kegg)." has been changed that "If the PARBs in the gut microbiota of golden snub-nosed monkeys spill into the human ecosystem, humans may get additional symptoms from the pathogenic pathways of pathogenic bacteria that tuberculosis, Staphylococcus, Streptococcus, Yersinia, pertussis, and Vibrio cholerae, and if human beings were infected by pathogenic pathways, humans may resistant to antifolate, platinum drugs, beta-lactam, vancomycin, and cationic antimicrobial peptide (https://www.genome.jp/kegg)" in lines 434-440 as the result also was showed in the response to comment (13). Thank you very much for your kind reminder.

Comment (15): Clarify the highlighted text.

Response to comment (15): After careful consideration, we have deleted the sentence that "Fortunately, the bacterial infectious disease and antimicrobial drug resistance to antibiotics of the PARBs in the gut of wild animals could be predicted by MAG technology through the KEGG system of humans.". Because this content of the sentence is similar to that of sentence that "KEGG pathways can define the complex interrelationships between genes and functions. If we want to predict the spillover risk of PARBs from wild animals to humans, the KEGG pathways of PARBs in antimicrobial drug resistance and bacterial infectious disease to humans should be studied.". We apologize for our mistake and thank you for your great comment.

Comment (16): Is that possible to mention the measures to be addressed?

The results confirmed our hypothesis, and the next steps are to formulate measures for health management and rescue of wild animals and obtain address emergency measures to prevent spillover of drug-resistant pathogens from wild animals.

Response to comment (16): We are so sorry that we did not mention what measures for health management of wild animals and what measures to prevent the spillover risk of PARBs from wild animals to humans. The sentence that "The results confirmed our hypothesis, and the next steps are to formulate measures for health management and rescue of wild animals and obtain address emergency measures to prevent spillover of drug-resistant pathogens from wild animals." has been replace by that " We recommend that the conflicts between wild animals and humans should be reduced, and the characteristics of ARGs, VFs and KEGG pathways of PARBs in wild animals are needed to be investigated under eco-environment coupling system of wild animals and humans, which could be used to predict the risks of PARBs to wild animals and find the proper antibiotics to treat the disease of wild animals and humans caused by PARBs." in the conclusions section to address the measures according to the research results. See lines 463-468.

We ensure that our revisions do not change the result of the manuscript. We hope that our modifications meet your requirements, but if there is anything inappropriate, please feel free to let us know.

Comment (17): Is that possible to provide a Conceptual Framework of the health risk assessment?

Response to comment (17): Thank you for your valuable comment. The Conceptual Framework is very important to clearly express the research content and significance of this manuscript, which has a great significance to promote the research results of this manuscript. The Conceptual Framework had been drawn, but it was added in cover letter. We fully accept this comment and the Conceptual Framework has been added in the revised manuscript as figure 2.

 Figure 2 Conceptual Framework of health risk assessment 

We tried our best to improve and revise the manuscript. These changes have not influenced the content and framework of the paper. The changes are not listed here but rather are marked in red in the revised paper.

We appreciate the valuable comments and suggestions of the Editors and Reviewers and hope that the corrections will meet with approval.

Once again, thank you very much for your comments and suggestions. If further changes are needed, please feel free to contact us.

Sincerely,

Dayong Li

980119lsc@163.com

Reviewer 3 Report

The article used golden snub-nosed monkeys as sentinel to investigate the health risk of human disturbance to wild animals and the spillover risk of PARBs from wild animals to humans, performing metagenomic analysis. The manuscript is interesting and deepens the correlation between wildlife-AMR dynamics-humans, not yet fully understood.

Title

I suggest to substitute "wild animal" with "golden snub-nosed monkeys".

Abstract

I suggest to briefly describe you analised faecal samples, perfoming metagenomic analysis, and then to describe the results.

lines 17-21: "The results........under HD (HD group) and wild habitat environments (W group), respectively". The sentence is too long and confusing. Please rephrase it.

line 23: "-28.5×10-3 and 125.8×10-3". What are you referring to with these numbers?  Please specify.

lines 28-29: "If these PARBs spill over to humans, new KEGG pathways of bacterial infectious disease were induced by HD from the PARBs...." I think the verb tense is incorrect.

Introduction

line 41-43: "Animal health, human health.............also determine human health". It looks redundant. It looks like you repeat the same thing in more sentences.

line 60: "If ARGs from HD enter the gut of wild animals". Substitute "from" with "associated".

line 66: "PARBs have been used as indicators to evaluate the health risk....". Normally non-pathogenic bacteria are used as indicators for AMR environmental pollution. Please rephrase it.

line 69: "pathogenic antibiotic-resistant bacteria (PARBs)". You already define PARBs (line 66).

lines 71-72: "However, the health risk of PARBs to wild animals and the spillover risk of PARBs from wild animals to humans have not been studied". I suggest not to be so categorical. You can say that only few studies focused on these aspects or that few data are available.

line 77-79: "Fortunately, nonhuman primates have been used as model animals to study diseases in humans, and their gut microbes are genetically similar to those carried by humans". The sentence looks out of place. Please remove or reformulate it.

line 83: "The results also showed that HD could make........." Which results? Do you mean "Peng et al also showed that HD could make...". Please rephrase it.

line 86-103: this paragraph is a little bit confusing. Additionally, I suggest to clearly specify the aims of your study (also with a list).

More in detail:

"We have used golden snub-nosed monkeys as sentinel animals to study the effect .......". How you wrote it, it looks like you obtain these data in the present study. I suggest to specify these results derived from previous studies, leading to the hypothesis "that HD will increase the health risks of PARBs to wild animals and the spillover risk of PARBs from wild animals to humans. To verify the hypothesis" you try to evaluate: 1)........., 2).......

line 98: KEGG. I suggest to briefly explain what KEGGs are and their importance. 

Research Methods

line 109: substitute "has" with "have" or change the sentence in "The Baihe National Nature Reserve has the highest density, largest and most representative population of Sichuan golden snub-nosed monkeys so far".

line 115-119: The paragraph is confusing. Are Xiaping ground monkeys and Sichuan golden snub-nosed monkeys different? As you stated, samples were collected only from Sichuan golden snub-nosed monkeys, so why you consider also Xiaping ground monkeys? Which group was separated from Xiaping ground monkeys? Please rephrase the paragraph.

line 123: "from groups W and HD", add "respectively" after "HD"

line 123: "and the fecal samples were randomly collected in the habitats of 123 groups W and HD". It sounds redundant, you already stated it in the sentence before.

line 126: remove "Then".

lines 128-131: Why some samples were brought to the laboratory and others to the Shanghai Magi Biological Company? Were they W or HD samples? How do you choose which sample was delivered to the first lab or the second one?

line 129: "laboratory". Which one? Please specify it.

DNA extraction (line 132): Which lab does perform it? I suggest to clearly specify the analysis performed in each lab.

line 158: add "respectively" after "groups".  "Representative samples": what are the features of representative samples? How do you choose them?

FIgure 1: Please write in full HD and W with acronyms in brackets in the figure's bibliography. What does light green indicate? 

line 216: what do you mean with "four attributes"?

Table 1: Why are some genes repeated many times in the table? Do you find the same gene on different contigs, which could be present in both W and HD?

Discussion

line 340: "The next steps" DO you mean you want to investigate these aspects in future studies?Please specify it. In this case I suggest to relocate the sentence in the conclusion chapter.

lines 352-353: the sentence is confusiong. Please rephrase it. 

lines 355: "types". Which types? Please specify.

lines 454: "our hypothesis". Specify it.

Author Response

Dear Reviewers:

Thank you for your letter and for the comments concerning our manuscript entitled "Human disturbance increases the health risks of golden snub-nosed monkeys and the spillover risks of pathogenic antibiotic-resistant bacteria from golden snub-nosed monkeys to humans" (ID: animals-2533952). Those comments were all valuable and very helpful for revising and improving our paper and provided important guiding significance for our research. We have studied the comments carefully and made corrections that we hope meet with approval. The revised portions are marked in red in the paper. The main corrections in the paper and the responses to the reviewer’s comments are as follows:

Reviewer #3:

Title

Comment (1): I suggest to substitute "wild animal" with "golden snub-nosed monkeys".  

Response to comment (1): Thank you for all of your valuable comments, we fully accept them.

The "wild animal" in the title has been substituted by "golden snub-nosed monkeys". In order to make the title more in line with the manuscript, elsewhere if needed, the "wild animal" also have been substituted by "golden snub-nosed monkeys".

Abstract

Comment (2): I suggest to briefly describe you analised faecal samples, perfoming metagenomic analysis, and then to describe the results.

Response to comment (2): We are sorry that the technology of metagenomic analysis did not be indicated the abstract section, and the sentence that "the metagenomic analysis was used to analyze the characteristics of PARBs of gut microbes in the golden snub-nosed monkeys" has been added in the abstract section. See lines 35 and 36. Thank you very much for your kind reminder.

Comment (3): lines 17-21: "The results........under HD (HD group) and wild habitat environments (W group), respectively". The sentence is too long and confusing. Please rephrase it.

Response to comment (3): We are so sorry for our confusing sentence. The sentence that "The results........under HD (HD group) and wild habitat environments (W group), respectively" has been rephrased by that " There were 18 and 5 kinds of PARBs in gut microbiota of the golden snub-nosed monkeys under HD (HD group) and wild habitat environments (W group), respectively. The total health risks of the PARBs to W group and HD group were -28.5×10-3 and 125.8×10-3, respectively. There were 12 and 16 kinds of KEGG pathways of human diseases in the PARBs of the W group and HD group, respectively, and the genes numbers of KEGG pathways of HD group were higher than that of W group. " in lines 41-47. 

Thank you very much for your kind reminder. We ensure that our revisions do not change the result of the manuscript. We hope that our modifications meet your requirements, but if there is anything inappropriate, please feel free to let us know.

Comment (4): line 23: "-28.5×10-3 and 125.8×10-3". What are you referring to with these numbers? Please specify.

Response to comment (4): The "-28.5×10-3 and 125.8×10-3 were the total health risks of the PARBs to W group and HD group, respectively. We are so sorry for our confusing sentence, and the "The total health risks of the W group and HD group were -28.5×10-3 and 125.8×10-3, respectively" has been changed to that "The total health risks of the PARBs to W group and HD group were -28.5×10-3 and 125.8×10-3, respectively. " in lines 43, 44 and 337. Thank you very much for your great comment.

Comments (5): lines 28-29: "If these PARBs spill over to humans, new KEGG pathways of bacterial infectious disease were induced by HD from the PARBs...." I think the verb tense is incorrect.

Response to comment (5): We apologize for our mistakes. The sentence that "If these PARBs spill over to humans, new KEGG pathways of bacterial infectious disease were induced by HD from the PARBs...." has been changed to that " If these PARBs spilled over from the golden snub-nosed monkeys to humans, the humans may be acquired symptoms of the pathogens of tubercle bacillus, Staphylococcus, Streptococcus, Yersinia, pertussis, and Vibrio cholera."(Lines 49-52). We hope that our modifications meet your requirements, but if there is anything inappropriate, please feel free to let us know.

Introduction

Comment (6): line 41-43: "Animal health, human health.............also determine human health". It looks redundant. It looks like you repeat the same thing in more sentences.

Response to comment (6): We are so sorry for our mistakes. The sentences that "Animal health, human health and environmental health are essentially interdependent. However, animal health and environmental health depend on human activities. Furthermore, animal health and environmental health also determine human health." has been changed to the sentence that " The wild animals health and their habitats health are affected by human activities". See lines 57 and 58. Thank you very much for your kind reminder.

Comment (7): line 60: "If ARGs from HD enter the gut of wild animals". Substitute "from" with "associated".

Response to comment (7): We apologize for our mistake, the "from" has been substituted with "associated" in line 73. Thank you very much.

Comment (8): line 66: "PARBs have been used as indicators to evaluate the health risk....". Normally non-pathogenic bacteria are used as indicators for AMR environmental pollution. Please rephrase it.

Response to comment (8): We fully accept your comment. The sentence that "PARBs could used to evaluate the health risk of HD to the ecological environment " has been changed to that "PARBs could be used to evaluate the health risk of HD to the ecological environment" (lines 79 and 80). Thank you very much.

Comment (9): line 69: "pathogenic antibiotic-resistant bacteria (PARBs)". You already define PARBs (line 66).

Response to comment (9): We apologize for carelessness, the "pathogenic antibiotic-resistant bacteria" has been deleted in the revised manuscript. Thank you very much for your kind reminder.

Comment (10): lines 71-72: "However, the health risk of PARBs to wild animals and the spillover risk of PARBs from wild animals to humans have not been studied". I suggest not to be so categorical. You can say that only few studies focused on these aspects or that few data are available.

Response to comment (10): We apologize for our mistake, the sentence that "However, the health risk of PARBs to wild animals and the spillover risk of PARBs from wild animals to humans have not been studied". has been changed to that "However, only few studies focused on the health risk of PARBs to golden snub-nosed monkeys and the spillover risk of PARBs from golden snub-nosed monkeys to humans.". See lines100-102. Thank you for your kind reminder.

Comment (11): line 77-79: "Fortunately, nonhuman primates have been used as model animals to study diseases in humans, and their gut microbes are genetically similar to those carried by humans". The sentence looks out of place. Please remove or reformulate it.

Response to comment (11): We fully accept your comment, and the sentence that "Fortunately, nonhuman primates have been used as model animals to study diseases in humans, and their gut microbes are genetically similar to those carried by humans" has been deleted in the revised manuscript. Thank you for your careful comment.

Comment (12): line 83: "The results also showed that HD could make........." Which results? Do you mean "Peng et al also showed that HD could make...". Please rephrase it.

Response to comment (12): We fully accept your comment. In order to makes it more clearly, the sentence that "The results also showed that HD could make........." has been changed to that " Peng et al showed that HD could make the gut microbiota of golden snub-nosed monkeys more similar to those of humans, and Yu et al indicated that the gut microbiota of golden snub-nosed monkeys could be used as a key indicator of their health. ". See lines lines 93-96.

Comment (13): line 86-103: this paragraph is a little bit confusing. Additionally, I suggest to clearly specify the aims of your study (also with a list).

More in detail: "We have used golden snub-nosed monkeys as sentinel animals to study the effect .......". How you wrote it, it looks like you obtain these data in the present study. I suggest to specify these results derived from previous studies, leading to the hypothesis "that HD will increase the health risks of PARBs to wild animals and the spillover risk of PARBs from wild animals to humans. To verify the hypothesis" you try to evaluate: 1)........., 2).......

Response to comment (13): We apologize for the confused sentences. The sentence that " In this study, the characteristics of PARBs in gut microbiota of golden snub-nosed monkeys were analyzed by metagenomic sequencing, which was used to study the effect of HD on health risks of PARBs to wild animals and the spillover risk of PARBs from wild animals to humans." was used to clearly specify the aims of our study in lines 103-106 of the revised manuscript.

We guarantee that this study is an independent experiment, the experimental data in this manuscript has never been published. The golden snub-nosed monkeys used in our previous published articles were lived in another national nature reserve in Yuannan province, so the sentence that "Our published studies have found that the relative abundances of ARGs and pathogenic bacteria in gut microbiota of golden snub-nosed monkeys in Yunnan were enhanced by HD" has been added in lines 96-98. In this study, the data in this manuscript was obtained by the metagenomic sequencing. In order to emphasize the independence of the data in this manuscript, the sentences that " In this study, the characteristics of PARBs in gut microbiota of golden snub-nosed monkeys were analyzed by metagenomic sequencing, which was used to study the effect of HD on health risks of PARBs to wild animals and the spillover risk of PARBs from wild animals to humans." was added in the lines 103-106.  

To specify these results derived from previous studies, the sentence that "We have used golden snub-nosed monkeys as sentinel animals to study the effect of HD on wild animals and found that the relative abundances of ARGs, VFs and drug resistant pathogens in gut microbes were enhanced by HD. We also found a risk of ARG and VF pollution in the habitat of golden snub-nosed monkeys" has been changed to that "Our published studies have found that the relative abundances of ARGs and pathogenic bacteria in gut microbiota of golden snub-nosed monkeys in Yunnan were enhanced by HD". See lines 96-98.

 To clearly specify the section, the sentence that "Thus, golden snub-nosed monkeys could be used as sentinel species to study the health risk of PARBs to wild animals and the spillover risk of PARBs from wild animals to humans." has been superseded by the sentence that "However, only few studies focused on the health risk of PARBs to golden snub-nosed monkeys and the spillover risk of PARBs from golden snub-nosed monkeys to humans."(Lines 100-102). As well as the last paragraph of the Introduction section has been changed in the revised manuscript, which was marked in red words.

We hope that our modifications meet your requirements, but if there is anything inappropriate, please feel free to let us know.

Comment (14): Line 98: KEGG. I suggest to briefly explain what KEGGs are and their importance.

Response to comment (14): We are so sorry for our mistake. The sentence that "KEGG pathways can define the complex interrelationships between genes and functions. If we want to predict the spillover risk of PARBs from wild animals to humans, the KEGG pathways of PARBs in antimicrobial drug resistance and bacterial infectious disease to humans should be studied." has been used to explain what was the KEGGs are and their importance in the lines 427-430 of Discussion section. To avoid ambiguity, "KEGG pathways in" has been deleted in the Introduction section.

Research Methods

Comment (15): line 109: substitute "has" with "have" or change the sentence in "The Baihe National Nature Reserve has the highest density, largest and most representative population of Sichuan golden snub-nosed monkeys so far".

Response to comment (15): We apologize for our carelessness. The sentence that "The Sichuan golden snub-nosed monkeys in Baihe National Nature Reserve have the highest density, largest and most representative population so far" has been changed to that "The Baihe National Nature Reserve has the highest density, largest and most representative population of golden snub-nosed monkeys so far. " in the revised manuscript. See lines 117 and 118.

Comment (16): line 115-119: The paragraph is confusing. Are Xiaping ground monkeys and Sichuan golden snub-nosed monkeys different? As you stated, samples were collected only from Sichuan golden snub-nosed monkeys, so why you consider also Xiaping ground monkeys? Which group was separated from Xiaping ground monkeys? Please rephrase the paragraph.

Response to comment (16): We apologize for our confusing sentence.

Our research objects are the golden snub-nosed monkeys of HD group and W group, all of them lived in the home ranges of Xiapingdi (Figure 2). The golden snub-nosed monkeys in some families in the home range of Xiapingdi have been severely disturbed by grazing[4], and they were fixed in a place of their home range in 2021 for ecotourism and scientific research, thus they have been disturbed by humans (HD group)[16]. The other golden snub-nosed monkey families in the home range of Xiapingdi live in mountains, the mountains are inaccessible and they never are disturbed by humans (W group). There were HD group and W group in the home range of Xiapingdi, so the HD group and W group of Xiapingdi golden snub-nosed monkeys were choose as the study subjects in this study.

   The paragraph has been rephrased in lines 116-130 of the revised manuscript. We hope the rephrased paragraph is more clearly. Thank you for your careful comment.

Comment (17): line 123: "from groups W and HD", add "respectively" after "HD".

Response to comment (17): We apologize for our mistake, the "respectively" has been added after the "HD" in the revised manuscript.

Comment (18): line 123: "and the fecal samples were randomly collected in the habitats of 123 groups W and HD". It sounds redundant, you already stated it in the sentence before.

Response to comment (18): We apologize for our mistake, the sentence that "and the fecal samples were randomly collected in the habitats of groups W and HD" and "A total of 50 and 53 fresh fecal samples from Sichuan golden snub-nosed monkeys from groups W and HD. " have been rephrased by the sentence that "A total of 50 and 53 fresh fecal samples of golden snub-nosed monkeys were randomly collected from groups W and HD, respectively.". in the revised manuscript. See lines 133 and 134. Thank you very much for your kind reminder.

Comment (19): line 126: remove "Then".

Response to comment (19): We fully accept your comment, the "Then" has been removed in the revised manuscript.

Comment (20): lines 128-131: Why some samples were brought to the laboratory and others to the Shanghai Magi Biological Company? Were they W or HD samples? How do you choose which sample was delivered to the first lab or the second one?

Response to comment (20): We apologize for the unclear sentences, the 16S rRNA and metagenomic sequencing of the gut microbes of W group and HD group was performed by the lab of Shanghai Magi Biological Company. Because the feces of the golden snub-nosed monkeys were difficult to obtain and require a lot of effort, a part of each feces was transported to our laboratory at China West Normal University in order to avoid the failure of the other part of each feces in the lab of Shanghai Magi Biological Company.

 Fortunately, the 16S rRNA and metagenomic sequencing was performed by the lab of Shanghai Magi Biological Company were very successful. So the sentence that "Some fecal samples were brought back to the laboratory for preliminary processing as soon as possible" was deleted to make the sentences clearer in the revised manuscript. And the sentences in this section have been rephrased by the sentence that " To ensure the freshness of the samples in groups W and HD, after all the fecal samples were collected, they were transported to the Shanghai Magi Biological Company on dry ice for 16S rRNA and metagenomic sequencing." in the revised manuscript. See lines 136-139.

Comment (21): line 129: "laboratory". Which one? Please specify it.

Response to comment (21): We apologize for our mistake, the 16S rRNA and metagenomic sequencing was performed by the lab of Shanghai Magi Biological Company were very successful. So this "laboratory" has been deleted based on your comment (20). Thank you for your kind reminder.

Comment (22): DNA extraction (line 132): Which lab does perform it? I suggest to clearly specify the analysis performed in each lab.

Response to comment (22): The DNA extraction was performed by the lab of Shanghai Magi Biological Company. We apologize for that the lab for DNA extraction was not indicated in this section. The sentence that "The DNA extraction was performed by the lab of Shanghai Magi Biological Company. " was added in lines 144 and 145 of the revised manuscript to specify the analysis of DNA extraction in revised manuscript.

Comment (23): line 158: add "respectively" after "groups". "Representative samples": what are the features of representative samples? How do you choose them?

Response to comment (23): Thank you for your kind reminder. The "respectively" has been added after "groups" in the revised manuscript.

The sentence that "The representative samples were selected according to β diversity of the gut microbiota to fully obtain all the information on the gut microbiota of golden snub-nosed monkeys. According to the results of the 16S rRNA sequencing, the 15 and 15 representative DNA samples of the gut microbiota were selected from 50 and 53 DNA samples for metagenomic sequencing analysis in the W and HD groups, respectively by the selection criterion of "most representative" in the micro-PITA method, and the representative samples were showed in Table S1." was used to explain how do you choose the representative samples in lines 161-167.

We are so sorry for the unclear sentences. We hope the rephrased paragraph is more clearly. Thank you for your great comment.

Comment (24): Figure 1: Please write in full HD and W with acronyms in brackets in the figure's bibliography. What does light green indicate?

Response to comment (24): Thank you for your kind reminder. The Figure 1 has been redrawn. The full HD and W with acronyms have been added in the redrawn figure. The light green indicated the home ranges of golden snub-nosed monkeys, and the sentence that "The home ranges of golden snub-nosed monkeys in Baihe National Nature Reserve were marked by the light green in Figure 1. " has been used to explain what the light green indicate. See lines 118-119.

Comment (25): line 216: what do you mean with "four attributes"?

Response to comment (25): Thank you for your kind reminder. The four attributes were the four independent variables in the factor analysis, they were the subtypes of ARGs, the gene numbers of ARGs, the subtypes of VFs, and the gene numbers of VFs in each PARB of W group or HD group. We are so sorry for the unclear words, the "four attributes" has been replaced by the "four independent variables". And the 2.5 section has been rephrased to explain what is the four independent variables in lines 228-230 in the revised manuscript.

Comment (26): Table 1: Why are some genes repeated many times in the table? Do you find the same gene on different contigs, which could be present in both W and HD?

Response to comment (26): Thank you for your kind reminder. Yes, there were same ARGs or VFs were found on different contigs. In table 1, the contigs of HD7_k97_107630_1, HD7_k97_208861_1, HD7_k97_226438_1, HD7_k97_309577_1, W34_k97_58465_1 and W34_k97_727054_1 were carried the same subtype of ARGs (ceoB).

If the contigs mumber is not "0" in W group or HD group, the contigs should present in W group and HD group. Some of the contigs were presented in both W group and HD group, such as the contigs of HD7_k97_107630_1, HD7_k97_226438_1, HD7_k97_334322_1, W34_k97_327771_1, W34_k97_590319_1, W42_k97_62472_1 and W44_k97_574071_1. Some of the contigs were presented in W group, they were HD50_k97_208870_1, HD7_k97_185188_1, HD7_k97_208861_1, HD7_k97_309577_1, HD7_k97_315010_1, HD7_k97_375936_1 and HD7_k97_401148_1. The other contigs were presented in HD group, they were W34_k97_1150608_1, W34_k97_132510_1, W34_k97_36958_1, W34_k97_551122_1, W34_k97_590319_1, W34_k97_713320_1, W34_k97_727054_1, W34_k97_939152_1, W42_k97_49042_1 and W7_k97_500225_1.

We apologize for our unclear expression in the table 1. The "Genes in W" and "Genes in HD" have been changed to "Contigs numbers in W" and "Contigs numbers in HD" in the revised table 1.

Discussion

Comment (27): line 340: "The next steps" Do you mean you want to investigate these aspects in future studies? Please specify it. In this case I suggest to relocate the sentence in the conclusion chapter.

Response to comment (27): We are so sorry for our mistake, and we fully accept your suggestion. The sentences that "The next steps are to increase the amount and sequencing depth of fecal samples of golden snub-nosed monkeys, identify the species of PARBs, and conduct more specific studies on their pathogenicity and drug resistance by pure culture of the PARBs." have been deleted in the discussion chapter. But the sentences have been relocated in the conclusion chapter. See lines 463-468.

Comment (28): lines 352-353: the sentence is confusiong. Please rephrase it.

Response to comment (28): We apologize for the confusing sentence. The sentence that "Our result also indicated that there was a potential risk from ARGs in golden snub-nosed monkeys (Figure 2C and 2E), which was supported by" has been changed to that "HD enhanced the potential risk of ARGs to golden snub-nosed monkeys, which was supported by previous results". See lines 381-383. As well as the first paragraph of the discussion section has been reorganized. We hope the rephrased paragraph is more clearly. Thank you for your great comment.

Comment (29): lines 355: "types". Which types? Please specify.

Response to comment (29): We apologize for this confusing words. The "types" was "classes of ARGs". We are so sorry for our mistake, the "types" has been changed to "classes of ARGs" in the revised manuscript. The sentence that "and HD significantly increased the relative abundances of ARGs in multidrug, tetracycline, MLS, and glycopeptide, as well as mupirocin and nucleoside types that were checked not only in the W group but also in the HD group." was the result of figure 2 in the result chapter and was move to the Results section. See lines 277-280.

Comment (30): lines 454: "our hypothesis". Specify it.

Response to comment (29): Thank you for your kind reminder. In this manuscript, the last paragraph of the Introduction section and Conclusions section have been reorganized, the what was the "our hypothesis" has been deleted. We hope the rephrased paragraph is more clearly and right. Thank you for your great comment.

We tried our best to improve and revise the manuscript. These changes have not influenced the content and framework of the paper. The changes are not listed here but rather are marked in red in the revised paper.

We appreciate the valuable comments and suggestions of the Editors and Reviewers and hope that the corrections will meet with approval.

Once again, thank you very much for your comments and suggestions. If further changes are needed, please feel free to contact us.

Sincerely,

Dayong Li

980119lsc@163.com

Round 2

Reviewer 1 Report

I declare that the authors have improved the manuscript according to my recommendations. However, authors should pay attention to the guidelines/instructions for authors. You are still not completely following them. 

For instance, regarding the abstract it is written "The abstract should be a single paragraph and should follow the style of structured abstracts, but without headings:"

You have added the headings in this last version. 

Author Response

Thank you for giving us the opportunity to revise the paper again. We apologize for our carelessness. The headings in the abstract section have been deleted in the revised manuscript this time. Thank you very much for your kind reminder. See lines 29 to 52.

Reviewer 2 Report

Dear Dayong Li (980119lsc@163.com),

Thank you for your comments and your hard work.

There are some technicalities to be improved, and English proofreading is needed to improve scientific writing. The health risk, and AMR and PARBs monitoring in golden snubnosed monkeys (as sentinel animals) is important for humans, wildlife, livestock, and environment, i.e., the One Health Approach. 

The title can be:

Health risk of pathogenic antibiotic-resistant bacteria for golden snub-nosed monkeys increased in the human disturbance habitat of Baihe National Nature Reserve of China.

1 Introduction

In this study, the characteristics of PARBs in gut microbiota of golden snub-nosed monkeys were analyzed by metagenomic sequencing, which was used to study the effect of HD on health risks of PARBs to wild animals and the spillover risk of PARBs from wild animals to humans. Our results showed that HD increased the pathogenicity of PARBs to wild animals and the PARBs in wild animals were resistant to lincosamide, aminoglycoside, and streptogramin antibiotics. If these PARBs spill over from the golden snub-nosed monkeys to humans, the humans may acquire symptoms of the pathogens of Tubercle bacillus, Staphylococcus, Streptococcus, Yersinia, pertussis, and Vibrio cholera which could guide to develop the management in the treatment of diseases in wild animals and emergency measures for spillover of pathogens to humans.

2. Research Methods

2.1 Study Subjects and Samples Collection

A total of 50 and 53 fresh fecal samples of golden snub-nosed monkeys were randomly collected from the W group and HD group, respectively. The feces were collected at 9 am and 5 pm every day. The methods for collection of fecal samples have been reported by Guo et al [21]. To ensure the freshness of the samples in groups W and HD, after all the fecal samples were collected, they were transported to the Shanghai Magi Biological Company on dry ice for 16S rRNA and metagenomic sequencing.

2.2 DNA extraction 2.3 16s rRNA and metagenomic sequencing analysis

2.4 Assembly of metagenome assembly genomes (MAGs)

2.5 Health risk of PARBs to the golden snub-nosed monkeys

The subtypes of ARGs, the numbers of ARGs, the subtypes of VFs, and the numbers of VFs in each PARB in the W group and HD group were four independent variables. We counted the subtypes of ARGs, the numbers of ARGs, the subtypes of VFs, and the numbers of VFs in each PARB in the W group and HD group, respectively, and the principal component features of the four independent variables were obtained by factor analysis. Then the total scores of the principal component features (Fi) were calculated by the Formula (2), the greater the score of the four independent variables was, the higher contribution of ARGs and VFs to risk of one PARB. The health risk of one PARB was affected by its abundance, so the health risk of one PARB was calculated by the Formula (3), and the total health risk of all PARBs in W group or HD group was calculated by the Formula (4)[37].

i is the number of component features in W group or HD group, Si is the score of the component features, Wi is the weight of Si and Fi is the total score of the PARB. HRi is the health risk of one PARB. and THR is the total health risk of all PARBs in the W group or the HD group. The greater the total health risk, the greater the health risk of PARBs to the golden snub-nosed monkeys[37]. Finally, KEGG pathways in antimicrobial drug resistance and bacterial infectious disease of PARBs were determined to assess the spillover risk of PARBs from golden snub-nosed monkeys to humans. The Conceptual Framework of health risk assessment to wild animals and humans was showed as figure 2.

2.6 Research techniques 3 Results

3.1 The effect of HD on ARGs and VFs in gut microbiota of golden snub-nosed monkeys

3.2 The effect of HD on the PARBs in gut microbiota of golden snub-nosed monkeys In the HD group, 12 species of MAGs were multidrug resistant class of PARBs in the HD group, while 4 species of MAGs were multidrug PARBs in the W group.

3.3 The effect of HD on health risk of golden snub-nosed monkeys

3.4 The effect of HD on spillover risk of the PARBs from golden snub-nosed monkeys to humans

Here you have a statistical analysis result and then can be discussed at point 4.

4 Discussion

At this point of the article, it is important to improve the discussion of the findings in the snub-nosed monkeys, taking into consideration that they are used for sentinel surveillance. The snub-nosed monkeys (as non-human primates), according to the literature cited from you in the introduction, are important to monitor:

The health of this endangered species.

The wildlife health.

Livestock health.

Environmental pollution by human disturbance.

Human health, as snub-nosed monkeys' gut microbiota, could be an indicator of human health.

Your findings in the snub-nosed monkeys' health are an indication of wildlife, livestock, environment, and human health in Baihe National Nature Reserve. The study is an indication that human, animal, and plant health are interdependent and bound to the health of the ecosystems in which they exist. https://www.woah.org/en/what-we-do/global-initiatives/one-health/

The study is an indication of the spillover of the PARBs in the human-animal-ecosystem but the spillover risk needs further study.

To be  improved

Author Response

Thank you for giving us the opportunity to revise the paper "Human disturbance increases the health risks of golden snub-nosed monkeys and the spillover risks of pathogenic antibiotic-resistant bacteria from golden snub-nosed monkeys to humans" (ID: animals-2533952) again. Those comments were all valuable and very helpful for improving our paper, especially the language in this manuscript. The revised portions are marked in red in the paper. The main corrections in the paper and the responses to the reviewer’s comments are as follows:

Comment (1): There are some technicalities to be improved, and English proofreading is needed to improve scientific writing.

Response to comment (1): Thank you very much for your kind reminder, your comments are important in improving the quality of this manuscript. We fully accept your comments, and the manuscript has been revised by the editing services at https://www.mdpi.com/authors/english (English editing ID: english-71522), the revisions have been marked in red. The invoice was showed as Figure 1. We apologize for our carelessness, Thank you for giving us the opportunity to revise the paper again.

Comment (2): The title can be:Health risk of pathogenic antibiotic-resistant bacteria for golden snub-nosed monkeys increased in the human disturbance habitat of Baihe National Nature Reserve of China. 

Response to comment (2): Thank you for your kind reminder, we fully accept your comment that the title of this manuscript should be modified. However, the gene numbers of mobile genetic element in the pathogenic antibiotic-resistant bacteria were analyzed in the manuscript, which showed that HD enhanced the transfer ability of the  pathogenic antibiotic-resistant bacteria. Then the gene numbers of KEGG pathways of human disease were also increased by HD. Our result indicated that the risk of the pathogenic antibiotic-resistant bacteria to humans may be increased by HD. The transfer risk of pathogenic antibiotic-resistant bacteria from golden snub-nosed monkeys to humans and the health risk of the pathogenic antibiotic-resistant bacteria for golden snub-nosed monkeys are equally important in this manuscript, so the title does not reflect only the health risks of the pathogenic antibiotic-resistant bacteria for golden snub-nosed monkeys. But, we have realized that spillover risk of the pathogenic antibiotic-resistant bacteria needs further study. So the phrases of "spillover risk" in the title has been changed to that "transfer risk". And the phrases pf "spillover risk" in other places of the manuscript have been changed to that "transfer risk". And the "spillover risk" in the Figure 2 (Conceptual Framework of health risk assessment) also has been changed to "transfer risk".

Figure 2 Conceptual Framework of health risk assessment 

We ensure that our revisions do not change the result of the manuscript. We hope that our modifications meet your requirements, but if there is anything inappropriate, please feel free to let us know.

Comment (3): 1 Introduction 

In this study, the characteristics of PARBs in gut microbiota of golden snub-nosed monkeys were analyzed by metagenomic sequencing, which was used to study the effect of HD on health risks of PARBs to wild animals and the spillover risk of PARBs from wild animals to humans. Our results showed that HD increased the pathogenicity of PARBs to wild animals and the PARBs in wild animals were resistant to lincosamide, aminoglycoside, and streptogramin antibiotics. If these PARBs spill over from the golden snub-nosed monkeys to humans, the humans may acquire symptoms of the pathogens of Tubercle bacillus, Staphylococcus, Streptococcus, Yersinia, pertussis, and Vibrio cholera which could guide to develop the management in the treatment of diseases in wild animals and emergency measures for spillover of pathogens to humans.

Response to comment (3): Thank you very much for your kind reminder, and we apologize for our carelessness. According to the editing result of the MDPI English Editing, these sentences that have been changed to that "In this study, the characteristics of PARBs in the gut microbiota of golden snub-nosed monkeys were analyzed via metagenomic sequencing, which was used to study the effect of HD on health risks of PARBs to wild animals and the spillover risk of PARBs from wild animals to humans. Our results showed that HD increased the pathogenicity of PARBs in wild animals, and the PARBs in wild animals were resistant to lincosamide, aminoglycoside, and streptogramin antibiotics. If these PARBs spill over from the golden snub-nosed monkeys to humans, humans may acquire symptoms of the pathogens of Tubercle bacillus, Staphylococcus, Streptococcus, Yersinia, pertussis, and Vibrio cholera, which could guide developing the management of the treatment of such diseases in wild animals and emergency measures for spillover of pathogens to humans.". See lines 98 to 107.

Comment (4): 2.1 Study Subjects and Samples Collection

A total of 50 and 53 fresh fecal samples of golden snub-nosed monkeys were randomly collected from the W group and HD group, respectively. The feces were collected at 9 am and 5 pm every day. The methods for collection of fecal samples have been reported by Guo et al [21]. To ensure the freshness of the samples in groups W and HD, after all the fecal samples were collected, they were transported to the Shanghai Magi Biological Company on dry ice for 16S rRNA and metagenomic sequencing.

Response to comment (4): Thank you very much for your kind reminder, and we apologize for our carelessness. According to the editing result of the MDPI English Editing, these sentences that have been changed to that "A total of 50 and 53 fresh fecal samples of golden snub-nosed monkeys were randomly collected from the W group and the HD group, respectively. The feces were collected at 9 am and 5 pm every day. The methods for collection of fecal samples have been reported by Guo et al. [21]. All fecal samples were transported to the Shanghai Magi Biological Company on dry ice for 16S rRNA and metagenomic sequencing after collection to ensure the freshness of the samples in groups W and HD.". See lines 128 to 134.

Comment (5): 2.2 DNA extraction

Response to comment (5): According to the editing result of the MDPI English Editing, these sentences in 2.2 section that have been changed to that "The FastDNA Spin Kit (MP Biomedicals, France) was used to extract DNA from 0.5 g of fresh fecal of the golden snub-nosed monkeys from the HD group the W group following the manufacturer’s instructions. Subsequently, spectrophotometric analysis (Nanodrop ND1000, USA) and 1.0% agarose gel electrophoresis were performed to assess the quality and concentration of isolated DNA. Finally, the isolated DNA was stored at -80 °C for further analysis. The DNA extraction was performed by the lab of Shanghai Magi Biological Company. ". See lines 136 to 142.

Comment (6): 2.3 16s rRNA and metagenomic sequencing analysis

Response to comment (6): The language in 2.3 and 2.4 sections have been modified by the MDPI English Editing, and the revised sentences have been showed in lines

Thank you very much.

Comment (7): 2.5 Health risk of PARBs to the golden snub-nosed monkeys

The subtypes of ARGs, the numbers of ARGs, the subtypes of VFs, and the numbers of VFs in each PARB in the W group and HD group were four independent variables. We counted the subtypes of ARGs, the numbers of ARGs, the subtypes of VFs, and the numbers of VFs in each PARB in the W group and HD group, respectively, and the principal component features of the four independent variables were obtained by factor analysis. Then the total scores of the principal component features (Fi) were calculated by the Formula (2), the greater the score of the four independent variables was, the higher contribution of ARGs and VFs to risk of one PARB. The health risk of one PARB was affected by its abundance, so the health risk of one PARB was calculated by the Formula (3), and the total health risk of all PARBs in W group or HD group was calculated by the Formula (4)[37]. 

i is the number of component features in W group or HD group, Si is the score of the component features, Wi is the weight of Si and Fi is the total score of the PARB. HRi is the health risk of one PARB. and THR is the total health risk of all PARBs in the W group or the HD group. The greater the total health risk, the greater the health risk of PARBs to the golden snub-nosed monkeys [37]. Finally, KEGG pathways in antimicrobial drug resistance and bacterial infectious disease of PARBs were determined to assess the spillover risk of PARBs from golden snub-nosed monkeys to humans. The Conceptual Framework of health risk assessment to wild animals and humans was showed as figure 2.

Response to comment (7): Thank you very much for your kind reminder, and we apologize for our carelessness. According to the editing result of the MDPI English Editing, these sentences in the 2.5 section that have been changed to that "The subtypes of ARGs, the numbers of ARGs, the subtypes of VFs, and the numbers of VFs in each PARB in the W group and HD group were four independent variables. We counted the subtypes of ARGs, the gene numbers of ARGs, the subtypes of VFs, and the gene numbers of VFs in each PARB in the W group and HD group, respectively, and the principal component features of the four independent variables were obtained via factor analysis. Then, the total scores of the principal component features were calculated using Formula (2). The greater the score of the four independent variables, the higher the contribution of ARGs and VFs to the risk of one PARB. The health risk of one PARB was affected by its abundance, so the health risk of one PARB was calculated using Formula (3), and the total health risk of all PARBs in the W group or HD group was calculated using Formula (4) [37]", " i is the number of component features in the W group or the HD group, Si is the score of the component features, Wi is the weight of Si and Fi is the total score of the PARB. HRi is the health risk of one PARB. and THR is the total health risk of all PARBs in the W group or the HD group. The greater the total health risk, the greater the health risk of PARBs to the golden snub-nosed monkeys [37]. ", and "Finally, KEGG pathways in antimicrobial drug resistance and bacterial infectious disease of PARBs were determined to assess the spillover risk of PARBs from golden snub-nosed monkeys to humans. The conceptual framework of the health risk assessment for wild animals and humans is shown in Figure 2. ". See lines 223 to 244.

We ensure that our revisions do not change the theme and meaning of the article. We hope that our modifications meet your requirements, but if there is anything inappropriate, please feel free to let us know.

Comment (8): 2.6 Research techniques

Response to comment (8): The revised sentences in the 2.6 sections were showed in lines 248 to 261.  

Comment (9): 3.1 The effect of HD on ARGs and VFs in gut microbiota of golden snub-nosed monkeys

Response to comment (9): The sentence that "The effect of HD on ARGs and VFs in gut microbiota of golden snub-nosed monkeys" has been changed to that " The effect of HD on ARGs and VFs in the gut microbiota of golden snub-nosed monkeys". Thank you very much for your kind reminder. See lines 263 and 264.

Comment (10): 3.2 The effect of HD on the PARBs in gut microbiota of golden snub-nosed monkeys

In the HD group, 12 species of MAGs were multidrug resistant class of PARBs in the HD group, while 4 species of MAGs were multidrug PARBs in the W group.

Response to comment (10): Thank you very much for your kind reminder. The sentence that "The effect of HD on the PARBs in gut microbiota of golden snub-nosed monkeys" has been changed to that "The effect of HD on the PARBs in the gut microbiota of golden snub-nosed monkeys". And the sentence that "In the HD group, 12 species of MAGs were multidrug resistant class of PARBs in the HD group, while 4 species of MAGs were multidrug PARBs in the W group." has been changed to that "In the HD group, 12 species of MAGs were multidrug PARBs, while four species of MAGs were multidrug PARBs in the W group. " in lines 300 and 301. We are so sorry for our carelessness.

Comment (11): 3.3 The effect of HD on health risk of golden snub-nosed monkeys 

Response to comment (11): The sentence that "The effect of HD on health risk of golden snub-nosed monkeys" has been changed to that "The effect of HD on health risk of the golden snub-nosed monkeys " in line 319 . Thank you very much.

Comment (12): 3.4 The effect of HD on spillover risk of the PARBs from golden snub-nosed monkeys to humans. Here you have a statistical analysis result and then can be discussed at point 4.

Response to comment (11): Thank you for your valuable comments. The T-test was used to analyse the gene numbers of KEGG pathways of antimicrobial drug resistance and bacterial infectious disease in PARBs, and result showed that the total gene numbers of KEGG pathways in bacterial infectious disease and antimicrobial drug resistance in the HD group were higher than in the W group (P<0.05; P<0.01). We did not discuss this part in depth, and we apologize for our carelessness. Thus, the sentences that "Our results indicated that HD significantly increased the ability of PARBs to cause disease in humans. Previous research has shown that the KEGG pathways of humans disease in the in gut microbiota of wild animals were enhanced by intervention, for the enrichment of pathogens in the gut of wild animals, which supported our result. As well as if the PARBs transfer to humans, the humans will be resistant to more classes of antibiotics. Thus, the gene numbers of MGEs and KEGG pathways numbers in antimicrobial drug resistance and bacterial infectious disease were enhanced by HD indicated that the transfer risks of PARBs from golden snub-nosed monkeys to humans were increased by HD." were used to discuss the statistical analysis the gene numbers of KEGG pathways of antimicrobial drug resistance and bacterial infectious disease in PARBs. See lines 437 to 446.

We hope that our modifications meet your requirements, but if there is anything inappropriate, please feel free to let us know. Thank you very much for your kind reminder.

4 Discussion:  At this point of the article, it is important to improve the discussion of the findings in the snub-nosed monkeys, taking into consideration that they are used for sentinel surveillance. The snub-nosed monkeys (as non-human primates), according to the literature cited from you in the introduction, are important to monitor: The health of this endangered species. The wildlife health. Livestock health. Environmental pollution by human disturbance. Human health, as snub-nosed monkeys' gut microbiota, could be an indicator of human health. Your findings in the snub-nosed monkeys' health are an indication of wildlife, livestock, environment, and human health in Baihe National Nature Reserve. The study is an indication that human, animal, and plant health are interdependent and bound to the health of the ecosystems in which they exist. The study is an indication of the spillover of the PARBs in the human-animal-ecosystem but the spillover risk needs further study.

Response to comment (14): Thank you for your great comments. We fully accept your comments that "it is important to improve the discussion of the findings in the snub-nosed monkeys, taking into consideration that they are used for sentinel surveillance. " and "The study is an indication of the spillover of the PARBs in the human-animal-ecosystem but the spillover risk needs further study.".

The sentences that "Our published studies have found that the relative abundances of ARGs and pathogenic bacteria in the gut microbiota of golden snub-nosed monkeys were enhanced by HD, and the potential risk of ARGs to golden snub-nosed monkeys was enhanced by HD, which supported our results. There is an interflow of genes among livestock, wild animals and humans, and humans can easily obtain ARGs and VFs from nonhuman primates, the risk of ARGs and VFs in the habitat of golden snub-nosed monkeys were enhanced by HD may be the cause that increased risk of ARGs in the golden snub-nosed monkeys. ", see lines 379 to 388.  and the sentences that "For disturbed and wild animals populations, deviations from ideal environmental conditions can be extremely harmful to health and may be associated with the occurrence and development of disease" were added to improve the discussion of the findings in the snub-nosed monkeys, taking into consideration that they are used for sentinel surveillance. See lines 402 to 404.

We have realized that our research is not enough to support spillover risk of the pathogenic antibiotic-resistant bacteria to humans and spillover risk of the pathogenic antibiotic-resistant bacteria to humans needs further study. So the phrases of "spillover risk" in the title has been changed to that "transfer risk". And the phrases pf "spillover risk" in other places of the manuscript have been changed to that "transfer risk". Beyond that, thank you for your valuable advice, which has pointed out the direction for our future research. The sentences that "However, due to the common limitation of incomplete MAGs [30], no MAGs were identified at the species level in our study. And the spillover of PARBs from wild animals to humans is related to the species of PARBs in wild animals, the characteristics of habitat environment and the availability of humans to the PARBs. Thus, in the future, the amount and sequencing depth of fecal samples of wild animals should be increased to identify the species of PARBs, the pathogenicity and drug resistance of the PARBs. What's more, the transfer path of PARBs from wild animals to humans and the collinearity of the PARBs and the microbiome of the humans hosts need to be studied under eco-environment coupling system of wild animals and humans. " were used to point out what we will to do to analyze the spillover risk of the pathogenic antibiotic-resistant bacteria to humans. See lines 458 to 467. We will look forward to your guidance and communication if we do these researches.

We hope that our modifications meet your requirements, but if there is anything inappropriate, please feel free to let us know.

Reviewer 3 Report

Dear Authors,

the manuscript has been deeply improved. Thank you for your efforts. Best regards.

Author Response

Comment (1): The manuscript has been deeply improved. Thank you for your efforts.

Response to comment (1): Thank you very much for your recognition to us, and thank you to give us great comments to improve our manuscript.